# CALIBRATED ADVERSARIAL REFINEMENT FOR STOCHASTIC SEMANTIC SEGMENTATION

## ABSTRACT

Ambiguities in images or unsystematic annotation can lead to multiple valid solutions in semantic segmentation. To learn a distribution over predictions, recent work has explored the use of probabilistic networks. However, these do not necessarily capture the empirical distribution accurately. In this work, we aim to learn a multimodal predictive distribution, where the empirical frequency of the sampled predictions closely reflects that of the corresponding labels in the training set. To this end, we propose a novel two-stage, cascaded strategy for calibrated adversarial refinement. In the first stage, we explicitly model the data with a categorical likelihood. In the second, we train an adversarial network to sample from it an arbitrary number of coherent predictions. The model can be used independently or integrated into any black-box segmentation framework to facilitate learning of calibrated stochastic mappings. We demonstrate the utility and versatility of the approach by attaining state-of-the-art results on the multigrader LIDC dataset and a modified Cityscapes dataset. In addition, we use a toy regression dataset to show that our framework is not confined to semantic segmentation, and the core design can be adapted to other tasks requiring learning a calibrated predictive distribution.

## 1 INTRODUCTION

Real-world datasets are often riddled with ambiguities, allowing for multiple valid solutions for a given input. These can emanate from an array of sources, such as ambiguous label space (Lee et al., 2016), sensor noise, occlusions, and inconsistencies or errors during manual data annotation. Despite this problem, the majority of the research encompassing semantic segmentation focuses on optimising models that assign a single prediction to each input image (Ronneberger et al., 2015; Jégou et al., 2017; Takikawa et al., 2019; Chen et al., 2017a;b; 2016a;b; 2015). These are often incapable of capturing the entire empirical distribution of outputs. Moreover, since they optimise for a one-fits-all solution, noisy labels can lead to incoherent predictions and therefore compromise their reliability (Lee et al., 2016).

Ideally, in such situations one would use a model that can sample multiple consistent hypotheses, capturing the different modalities of the ground truth distribution, and leverage uncertainty information to identify potential errors in each. Further, the sampled predictions should accurately reflect the occurrence frequencies of the labels in the training set; that is, the predictive distribution should be calibrated (Guo et al., 2017; Kull et al., 2019). Such a system would be particularly useful for hypothesis-driven reasoning in human-in-the-loop semi-automatic settings. For instance, large scale manual annotation of segmentation map is very labour-intensive—each label in the Cityscapes dataset takes on average 1.5 hours to annotate (Cordts et al., 2016). Alternatively, having a human operator manually selecting from a set of automatically generated label proposals could accelerate this process dramatically. In addition, combining uncertainty estimates with sampling of self-consistent labels, can be used to focus the annotator's attention to ambiguous regions, where errors are likely to occur, thereby improving safety.

Several approaches have been proposed to capture label multimodality in image-to-image translation tasks (Huang et al., 2018; Lee et al., 2018; Zhu et al., 2017a; Bao et al., 2017; Zhang, 2018), with only a few of them applied on stochastic semantic segmentation (Kohl et al., 2018; 2019; Baumgartner et al., 2019; Hu et al., 2019; Kamnitsas et al., 2017; Rupprecht et al., 2017; Bhattacharyya et al., 2018). These methods have the capacity to learn a diverse set of labels for each input, however, they

are either limited to a fixed number of samples (Kamnitsas et al., 2017; Rupprecht et al., 2017), return uncalibrated predictions, or do not account for uncertainty.

In this work, we tackle all three challenges by introducing a two-stage cascaded strategy. In the first stage we estimate pixelwise class probabilities and in the second, we sample confident predictions, calibrated relatively to the distribution predicted in the first stage. This allows us to obtain both uncertainty estimates as well as self-consistent label proposals. The key contributions are [1]:

- We propose a novel cascaded architecture that constructively combines explicit likelihood modelling with adversarial refinement to sample an arbitrary number of confident, and self-consistent predictions given an input image.
- We introduce a novel loss term that facilitates learning of calibrated stochastic mappings when using adversarial neural networks. To our knowledge this is the first work to do so.
- The proposed model can be trained independently or used to augment any pretrained black-box semantic segmentation model, endowing it with a multimodal predictive distribution.

## 2 RELATED WORK

Straightforward strategies towards learning multiple predictions include ensembling (Lakshminarayanan et al., 2017; Kamnitsas et al., 2017) or using multiple prediction heads (Rupprecht et al., 2017). Even though these approaches can capture a diverse set of sampled predictions, they are limited to only a fixed number of samples. Alternatively, a probability distribution over the outputs can be induced by activating dropout during test time (Gal and Ghahramani, 2016a). This method does offer useful uncertainty estimates over the pixel-space (Mukhoti and Gal, 2018), however, Isola et al. (2017) and Kohl et al. (2018) have demonstrated that it introduces only minor stochasticity in the output and returns incoherent samples.

Bhattacharyya et al. (2018) identify the maximum likelihood learning objective as the cause for this phenomenon in dropout Bayesian neural networks (Gal and Ghahramani, 2016b). They postulate that under cross entropy optimisation, all sampled models are forced to explain all the data, and thereby converge to the mean solution. To counter that, they propose to replace the cross entropy with and adversarial loss term parametrising a synthetic likelihood (Rosca et al., 2017), thereby making it conducive to multimodality. In contrast to this method, our approach is simpler to implement as it is not cast in the framework of variational Bayes, which requires the specification of weight priors and variational distribution family.

Kohl et al. (2018) take an orthogonal approach in combining a U-Net (Ronneberger et al., 2015) with a conditional variational autoencoder (cVAE) (Kingma and Welling, 2013) to learn a distribution over semantic labels. In Kohl et al. (2019) and Baumgartner et al. (2019) the authors build on Kohl et al. (2018) to improve the diversity of the samples by modelling the data on several scales of the image resolution. Nonetheless, these methods do not explicitly calibrate the predictive distribution in the pixel-space, and consequently do not provide reliable aleatoric uncertainty estimates (Kendall and Gal, 2017; Choi et al., 2018; Gustafsson et al., 2019). Hu et al. (2019) address this shortcoming by using the intergrader variability as additional supervision. A major limitation of this approach is the requirement of a-priori knowledge of all the modalities of the data distribution. For many real-world datasets, however, this information is not readily available.

In the more general domain of image-to-image translation, alternative methods employ hybrid models that use adversarially trained cVAEs (Zhu et al., 2017a; Bao et al., 2017) to learn a distribution over a latent code, capturing multimodality, in order to sample diverse and coherent predictions. A common hurdle in conditional generative adversarial network (cGAN) approaches is that simply incorporating a noise vector as an additional input often results in mode collapse. This occurs due to the lack of regularisation between the noise input and generator output, allowing the generator to learn to ignore the noise vector (Isola et al., 2017). This issue is commonly resolved by using supplementary cycle-consistency losses (Huang et al., 2018; Lee et al., 2018; Zhu et al., 2017a; Bao et al., 2017), as proposed by Zhu et al. (2017b) or with alternative regularisation losses on the generator (Yang et al., 2018). However, none of these methods explicitly address the challenge of calibrating the predictive distribution.

---

[1]Code is publicly available at <URL OMITTED FOR ANONYMITY>

## 3 PRELIMINARIES

Given an input image $x \in \mathbb{R}^{H \times W \times C}$, semantic segmentation refers to the task of predicting a pixel-wise class label $y \in \{1, \ldots, K\}^{H \times W}$. For a dataset of $N$ image and label pairs, $\mathcal{D} = \{x_i, y_i\}_{i=1}^{N}$, the conditional distribution $p_{\mathcal{D}}(y \mid x)$ can be explicitly modelled through a likelihood $q_{\theta}(y \mid x)$, parametrised by a convolutional neural network $F$ with weights $\theta$, and activated by a softmax function (Ronneberger et al., 2015; Jégou et al., 2017). One simple, yet effective way to learn the class probabilities is to express $y \in \{0, 1\}^{H \times W \times K}$ as a one-hot encoded label, and set $q_{\theta}$ as a pixelwise factorised categorical distribution, given by:

$$q_{\theta}(y \mid x) = \prod_{i}^{H} \prod_{j}^{W} \prod_{k}^{K} F_{\theta}(x)_{i,j,k}^{y_{i,j,k}}. \tag{1}$$

The parameters $\theta$ are then optimised by minimising the cross entropy between $p_{\mathcal{D}}$ and $q_{\theta}$, defined as:

$$\mathcal{L}_{\text{ce}}(\mathcal{D}, \theta) = -\mathbb{E}_{p_{\mathcal{D}}(x,y)}[\log q_{\theta}(y \mid x)]. \tag{2}$$

When trained with Eq. (2), $F_{\theta}$ learns an approximation of $\mathbb{E}_{p_{\mathcal{D}}}[y \mid x]$ (Bishop, 2006) that captures the pixelwise class probabilities over the label corresponding to a given input. This accommodates the quantification of the amount of noise inherent to the data, also referred to as aleatoric uncertainty, which can be obtained by computing the entropy of the output of $F_{\theta}$, $\mathbb{H}(F_{\theta}(x))$ (Kendall and Gal, 2017). Further, the final segmentation map is typically obtained by applying the $\text{argmax}$ function along the class dimension of the predicted probabilities. However, due to its deterministic nature, this approach is unable to produce multiple alternative predictions for the same input image. On the other hand, direct sampling from $q_{\theta}$ yields incoherent semantic maps, as for noisy datasets, maximising the factorised likelihood results in unconfident predictions in regions of inter-label inconsistencies, e. g. fuzzy object boundaries, exemplified later in the model overview diagram in Fig. 1.

The aforementioned limitations can be partially addressed by adapting the framework of generative adversarial networks (GANs) (Goodfellow et al., 2014) to the context of conditional semantic segmentation, as proposed by Luc et al. (2016). Formally, this involves training a binary discriminator network $D$ to optimally distinguish between ground truth and predictions, while concurrently training a generative network $G$ to maximise the probability that prediction samples $G(x)$ are perceived as real by $D$. Importantly, in contrast to explicit pixelwise likelihood maximisation, the adversarial setup learns an implicit sampler through $G$, capable of modelling the joint pixel configuration of the synthesised labels, and capturing both local and global consistencies present in the ground truth.

In practice, the generator loss is often complemented with the pixelwise loss from Eq. (2) to improve training stability and prediction quality (Luc et al., 2016; Ghafoorian et al., 2018; Samson et al., 2019). However, we argue that the two objective functions are not well aligned in the presence of noisy data. While the categorical cross entropy optimises for a single solution for each input $x$, thus encouraging high entropy in $q_{\theta}(y \mid x)$ within noisy regions of the data and calibrating the predictive distribution, the adversarial term allows multiple solutions while optimising for low entropy, label-like output. Therefore combining these losses in an additive manner, and enforcing them on the same set of parameters can be suboptimal. This issue can be mitigated to some extent by a scheduled downscaling of $\mathcal{L}_{\text{ce}}$, however, the residual conflict between the two losses adversely affects the optimisation process.

## 4 METHOD

### 4.1 CALIBRATED ADVERSARIAL REFINEMENT

In this work, we propose to avert potential conflict between the cross entropy and adversarial losses by decoupling them in a two-stage cascaded architecture consisting of a *calibration network* $F_{\theta}$ optimised with $\mathcal{L}_{\text{ce}}$ from Eq. (2), the output of which is fed to a *refinement network* $G_{\phi}$. $G_{\phi}$ is then optimised with an adversarial loss term, parametrised by an auxiliary discriminator $D_{\psi}$ trained with a binary cross entropy loss. To account for the multimodality in the labels, we additionally condition the refinement network on an extraneous noise variable $\epsilon \sim \mathcal{N}(0, 1)$. In practice, we also condition the refinement network and the discriminator on input $x$, however, we do not show this explicitly for notational convenience. More formally, using the non-saturated version of the

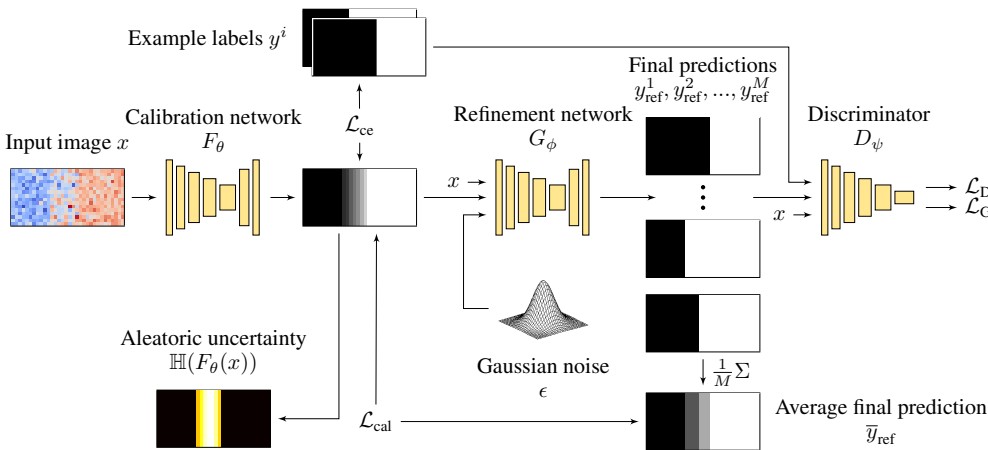

Figure 1: Model overview with an illustrative example where red and blue pixels are vertically segmented. A fuzzy boundary in the input image $x$ allows for multiple valid ground truth labels $y^i$. First, the calibration network maps the input to a calibrated pixelwise distribution over the labels. This is then fed into the refinement network which samples an arbitrary number of diverse, crisp label proposals $y^1_{\text{ref}}, \ldots, y^M_{\text{ref}}$. To ensure calibration, the average of the final predictions is matched with the calibration target from the first stage through the $\mathcal{L}_{\text{cal}}$ loss. Additionally, the aleatoric uncertainty can be readily extracted from the calibration target, e. g. by computing the entropy $\mathbb{H}(F_\theta(x))$.

adversarial loss (Goodfellow et al., 2014), the objectives for the refinement and discriminator networks respectively are given by:

$$\mathcal{L}_{\text{adv}}(\mathcal{D}, \theta, \phi) = -\mathbb{E}_{p_{\mathcal{D}}(x,y), p(\epsilon)}[\log D_\psi(G_\phi(F_\theta(x), \epsilon))], \tag{3}$$

$$\mathcal{L}_{\text{D}}(\mathcal{D}, \theta, \phi, \psi) = -\mathbb{E}_{p_{\mathcal{D}}(x,y), p(\epsilon)}[\log(1 - D_\psi(G_\phi(F_\theta(x), \epsilon))) + \log D_\psi(y)]. \tag{4}$$

To calibrate the predictive distribution, we impose diversity regularisation on $G_\phi$ by introducing a novel loss term that encourages the sample average $\overline{G}_\phi(F_\theta(x)) := \mathbb{E}_{p(\epsilon)}[G_\phi(F_\theta(x), \epsilon)]$ to match the class probabilities predicted by $F_\theta(x)$. Here, $\overline{G}_\phi(F_\theta(x))$ serves as an approximation of the implicit predictive distribution of the refinement network. To this end, we define an auxiliary fully-factorised categorical likelihood $q_\phi$ as:

$$q_\phi(y \mid F_\theta(x)) = \prod_i^H \prod_j^W \prod_k^K \overline{G}_\phi(F_\theta(x))^{y_{i,j,k}}_{i,j,k}, \tag{5}$$

where $\phi$ is optimised by minimising the Kullback-Leibler divergence $\text{KL}(q_\phi \parallel q_\theta)$.[2] Since both $q_\phi$ and $q_\theta$ are categorical distributions, the divergence can be computed exactly. We coin this loss term as the *calibration loss*, defined as:

$$\mathcal{L}_{\text{cal}}(\mathcal{D}, \theta, \phi) = \mathbb{E}_{p_{\mathcal{D}}}\left[\mathbb{E}_{q_\phi}[\log q_\phi(y \mid F_\theta(x)) - \log q_\theta(y \mid x)]\right]. \tag{6}$$

Since $\mathcal{L}_{\text{cal}}$ optimises through $\overline{G}_\phi(F_\theta(x))$ rather than a single sampled prediction, the model is not restricted to learning a single mode-averaging solution for each input $x$, and is therefore more compatible with $\mathcal{L}_{\text{adv}}$. The total loss for the refinement network then becomes:

$$\mathcal{L}_{\text{G}}(\mathcal{D}, \theta, \phi) = \mathcal{L}_{\text{adv}}(\mathcal{D}, \theta, \phi) + \lambda \mathcal{L}_{\text{cal}}(\mathcal{D}, \theta, \phi), \tag{7}$$

where $\lambda \geq 0$ is an adjustable hyperparameter. Fig. 1 shows the interplay of $F_\theta$, $G_\phi$ and $D_\psi$ and the corresponding loss terms.

Intuitively, the calibration network $F_\theta$ serves three main purposes. It accommodates the extraction of sample-free aleatoric uncertainty maps. It provides $G_\phi$ with an augmented representation of $x$ enclosing probabilistic information about $y$. Finally, it sets a calibration target used by $\mathcal{L}_{\text{cal}}$ to regularise the

---

[2]The choice of divergence is heuristically motivated and can be changed to fit different use-case requirements. We delegate theoretical and experimental analysis of other divergences to future work.

predictive distribution of $G_\phi$, in a cycle-consistent manner (Zhu et al., 2017b). The refinement network can then be interpreted as a stochastic sampler, modelling the inter-pixel dependencies to draw self-consistent samples from the explicit likelihood provided by the calibration network. Thus both the pixelwise class probability and object coherency are preserved. This approach leads to improved mode coverage and training stability, and increased convergence speed, as demonstrated in Section 5.

### 4.2 Practical considerations

An important consequence of the loss decomposition is that the weights of $F_\theta$ can be kept fixed, while the adversarial pair $G_\phi$ and $D_\psi$ are being trained. This allows $F_\theta$ to be pretrained in isolation, consequently lowering the overall peak computational burden and improving training stability (see Algorithms 1 and 2 in Appendix A.1 for an outline of the training and inference procedures). Further, computing $\mathcal{L}_{\text{cal}}$ requires a Monte Carlo estimation of $\overline{G}_\phi(F_\theta(x))$, where the quality of the loss feedback increases with the sample count. However, modern deep learning frameworks allow for the samples to be subsumed in the batch dimension, and can therefore be efficiently computed on GPUs. We also note that on noisy data points, the trained model does not require to be provided with all solutions for a given input. Instead, during training a random input-label pair is sampled, and the model automatically learns to identify cases where the input is ambiguous. Finally, our method can augment any existing black-box model $B$ for semantic segmentation, furnishing it with a multimodal predictive distribution. This can be done by conditioning $F_\theta$ on the output of $B$, which we demonstrate in Section 5.2.2.

## 5 Experiments

### 5.1 1D bimodal regression

We give intuitive insight into the mechanics of the proposed calibration loss by designing and experimenting on a simple one-dimensional regression task. To create the dataset, an input $x \in [0, 1]$ is mapped to $y \in \mathbb{R}$ as follows:

$$
y = \begin{cases} 0.5 - b + \epsilon, & x \in [0, 0.4) \\ (-1)^b(-1.25x + 1) + \epsilon, & x \in [0.4, 0.8) \\ \epsilon, & x \in [0.8, 1] \end{cases} \tag{8}
$$

where $b \sim \text{Bernoulli}(\pi)$ and $\epsilon \sim \mathcal{N}(0, \sigma)$. We generate 9 different scenarios by varying the degree of mode selection probability $\pi \in \{0.5, 0.6, 0.9\}$ and the mode noise $\sigma \in \{0.01, 0.02, 0.03\}$.

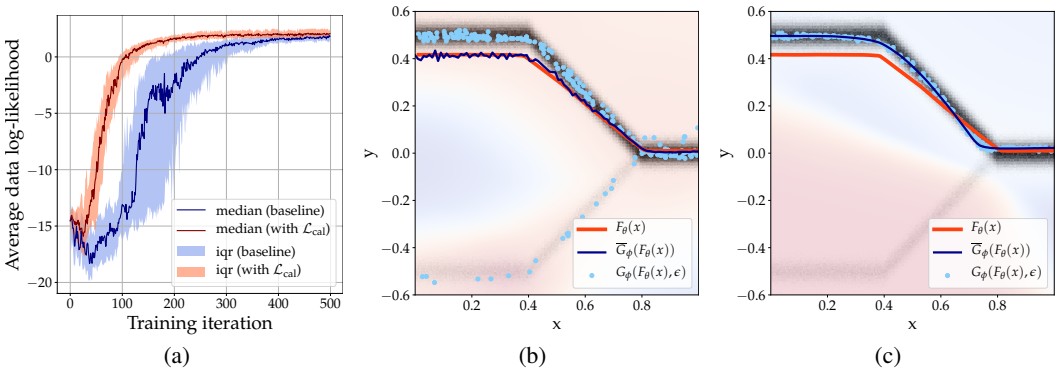

(a)   (b)   (c)

Figure 2: **(a)** Median and interquartile range (iqr) over the data log-likelihood, averaged over all 9×5×2 experiments. **(b)** High bias and noise configuration ($\pi = 0.9$, $\sigma = 0.03$) with calibration loss. The ground truth target is shown as black dots and the predicted samples as light blue dots. The predictions average in dark blue matches the calibration target in red. The discriminator output is shown in the background in shades of red (real) and blue (fake). **(c)** The same experiment configuration but without the proposed calibration loss, resulting in a mode collapse.

For every data configuration, we use a 4-layer MLP for each of $F_\theta$, $G_\phi$ and $D_\psi$, and train with and without calibration loss by setting the coefficient $\lambda$ in Eq. (7) to 1 or 0, respectively. All runs are trained with a learning rate of $1\mathrm{e}{-4}$, and each experiment is repeated five times. Note that unlike the categorical likelihood used in semantic segmentation tasks, we employ a Gaussian likelihood with fixed scale of 1. This changes the formulation of both Eqs. (2) and (6) to mean squared error losses between ground truth labels $y$ and predictions $\hat{y}$ for $\mathcal{L}_{\mathrm{ce}}$, and between the output of the calibration network $F_\theta(x)$ and the average of multiple predictions $\overline{G}_\phi(F_\theta(x))$ for $\mathcal{L}_{\mathrm{cal}}$ (see Appendix A.2).

The results, depicted in Fig. 2, illustrate that when using calibration loss, the optimisation process shows improved stability, converges faster and results in better calibrated predictions, in comparison to the non-regularised baseline. The effect is more pronounced in data configurations with higher bias. Further plots of the individual experiments are presented in Appendix B.1.

## 5.2 Stochastic semantic segmentation

In this section we examine the capacity of our model to learn shape and class multimodality in real-world segmentation datasets. We begin by sharing essential implementation details below.

**Network architectures** For the calibration network $F_\theta$, we use the encoder-decoder architecture from SegNet (Badrinarayanan et al., 2017). For $G_\phi$, we designed a U-Net-style (Ronneberger et al., 2015) architecture with 4 down- and upsampling blocks, each consisting of a convolutional layer, followed by a batch normalisation layer (Ioffe and Szegedy, 2015), a leaky ReLU activation, and a dropout layer (Srivastava et al., 2014) with 0.1 dropout probability. We use a base number of 32 channels, doubled or halved at every down- and upsampling transition. To propagate the sampled noise vector to the output, we inject it into every upsampling block of the network in an affine manner. To do so, we project the noise vector using two fully connected layers into scale and residual matrices, with the same number of channels as the feature maps at the points of injection, and use these matrices to adjust the channel-wise mean and variance of the activations. This is similar to the mechanism used for adaptive instance normalisation (Huang and Belongie, 2017). We base the architecture for $D_\psi$ on that used in DC-GAN (Radford et al., 2015) except that we remove batch normalisation. Any deviations from this setup are described in the corresponding sections.

**Training details** We utilise the Adam optimiser (Kingma and Ba, 2014) with an initial learning rate of $2\mathrm{e}{-4}$ for $F_\theta$ and $G_\phi$, and $1\mathrm{e}{-5}$ for $D_\psi$. The learning rates are linearly decayed over time and we perform scheduled updates to train the networks. Additionally, the discriminator loss is regularised by using the $R_1$ zero-centered gradient penalty term (Mescheder et al., 2018). For a detailed list of hyperparameter values, see Appendix A.1.

**Metrics** Following Kohl et al. (2018; 2019), Huang et al. (2018) and Baumgartner et al. (2019), we use the Generalised Energy Distance (GED) (Székely and Rizzo, 2013) metric:

$$D^2_{\mathrm{GED}}(p_\mathcal{D}, q_\phi) = 2\mathbb{E}_{s\sim q_\phi, y\sim p_\mathcal{D}}[d(s,y)] - \mathbb{E}_{s,s'\sim q_\phi}[d(s,s')] - \mathbb{E}_{y,y'\sim p_\mathcal{D}}[d(y,y')], \qquad (9)$$

where $d(s,y) = 1 - \mathrm{IoU}(s,y)$. As an additional metric, we follow Kohl et al. (2019) in using the Hungarian-matched IoU (HM-IoU) (Kuhn, 2005). In contrast to GED, which naively computes diversity as 1-IoU between all possible pairs of ground truth or sampled predictions, HM-IoU finds the optimal 1:1 matching between all labels and predictions, and therefore is more representative of how well the learnt predictive distribution fits the ground truth.

All experiments are performed in triplicate and we report results as mean and standard deviation. Further details regarding the exact implementation of the GED and HM-IoU metrics for each experiment can be found in Appendix A.3.

### 5.2.1 Learning shape diversity on the LIDC dataset

The Lung Image Database Consortium (LIDC) (Armato III et al., 2011) dataset consists of 1018 thoracic CT scans from 1010 lung cancer patients, graded independently by four expert annotators. We use the 180×180 crops from the preprocessed version of the LIDC dataset used and described in Kohl et al. (2018). The dataset is split in 8882, 1996 and 1992 images in the training, validation and test sets respectively. All models are trained on lesion-centered 128×128 crops where at least one of the four annotations indicates a lesion. The final evaluation is performed on the provided test set.

Table 1: Mean GED and HM-IoU scores on LIDC. Top section: approaches using the original data splits defined by Kohl et al. (2018), which we also adhere to; middle: approaches using random data splits; bottom: the $\mathcal{L}_{ce}$-regularised baseline and our $\mathcal{L}_{cal}$-regularised cGAN. The three central columns show the GED score computed with 16, 50 and 100 samples, respectively. The rightmost column shows the HM-IoU score, computed with 16 samples. The arrows ↑ and ↓ indicate if higher or lower score is better.

| Method | GED ↓ (16) | GED ↓ (50) | GED ↓ (100) | HM-IoU ↑ (16) |
|---|---|---|---|---|
| Kohl et al. (2018) | $0.320 \pm 0.030$ | — | $0.252 \pm$ N/A[1] | $0.500 \pm 0.030$ |
| Kohl et al. (2019) | $0.270 \pm 0.010$ | — | — | $0.530 \pm 0.010$ |
| Hu et al. (2019) | — | $0.267 \pm 0.012$ | — | — |
| Baumgartner et al. (2019) | — | — | $\mathbf{0.224 \pm}$**N/A**[2] | — |
| cGAN+$\mathcal{L}_{ce}$ | $0.639 \pm 0.002$ | — | — | $0.477 \pm 0.004$ |
| cGAN+$\mathcal{L}_{cal}$ | $\mathbf{0.264 \pm 0.002}$ | $\mathbf{0.248 \pm 0.004}$ | $0.243 \pm 0.004$[2] | $\mathbf{0.592 \pm 0.005}$ |

[1] This score is taken from Baumgartner et al. (2019).
[2] Note that when following the data split methodology used in Baumgartner et al. (2019) and computing the GED (100) metric, we achieve a score of 0.228 instead of 0.243 (see Appendix A.3).

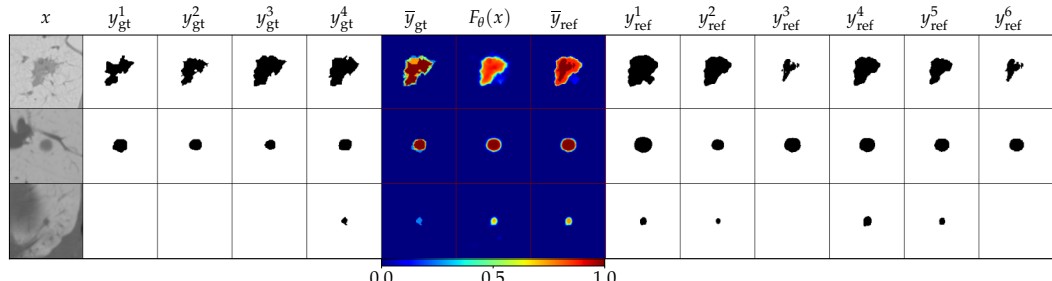

Figure 3: LIDC validation samples. From left to right: an input image $x$, followed by the four ground truth annotations $y_{gt}^1 \ldots y_{gt}^4$, the mean of the labels $\overline{y}_{gt}$, the output of the calibration network $F_\theta(x)$, the mean of the six refinement network samples $\overline{y}_{ref}$, shown in columns $y_{ref}^1 \ldots y_{ref}^6$.

To assess the accuracy and diversity of samples generated by our model we examine how well the learnt conditional distribution captures the shape diversity of lesion segmentations in the dataset. We pretrain $F_\theta(x)$ with $\mathcal{L}_{ce}$ in isolation, fix its weights, and then train $G_\phi$ with $\mathcal{L}_G$, where we estimate $\mathcal{L}_{cal}$ with 20 samples from $G_\phi$. Further experiments with varying sample size are disclosed in Table 3 in Appendix B.2. As a control experiment, we train using the same architecture but replace $\mathcal{L}_{cal}$ in the refinement network loss function with a cross entropy loss $\mathcal{L}_{ce}$, as used in Luc et al. (2016).

The $\mathcal{L}_{cal}$-regularised model performs on par with other state-of-the-art methods w. r. t. the GED score, and outperforms them on the HM-IoU score (only available for Kohl et al. (2018; 2019)). Numerical results are summarised in Table 1. The diversity and fidelity of sampled predictions are illustrated in Fig. 3. In contrast, the $\mathcal{L}_{ce}$-regularised baseline collapses the predictive distribution, showing no perceptible diversity in the samples (see Fig. 10b in Appendix B.2.3), which results in a stark increase in the mean GED score, and decrease in the HM-IoU score, as shown in the bottom section of Table 1.

### 5.2.2 Learning a calibrated distribution on a multimodal Cityscapes dataset

The Cityscapes dataset contains 1024×2048 RGB images of urban scenes, and their corresponding segmentation maps. It consists of 2975 training, 500 validation and 1525 test images. Following Kohl et al. (2018), we use the version of the Cityscapes dataset with 19 semantic classes and downsampled images and segmentation maps at a spatial resolution of 256×512. They establish controlled multi-modality by augmenting the dataset with 5 new classes: *sidewalk2*, *person2*, *car2*, *vegetation2* and *road2*, introduced by flipping their original counterparts with probabilities $8/17$, $7/17$, $6/17$, $5/17$ and $4/17$, respectively (see Fig. 4a). Following Kohl et al. (2018), we report results on the validation set.

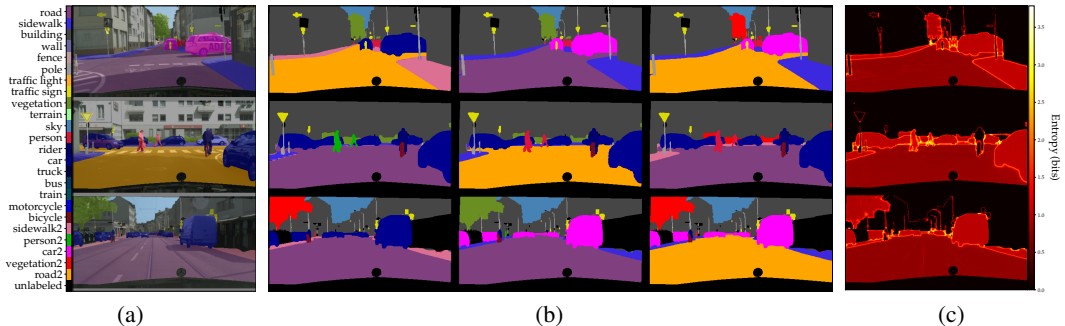

Figure 4: **(a)** Input images overlaid with the corresponding labels. **(b)** Samples obtained from the refinement network. **(c)** Aleatoric uncertainty computed as the entropy of the calibration output.

To demonstrate that our approach can be easily integrated on top of any existing black-box segmentation model $B$, we employ the network from TensorFlow DeepLab Model Zoo (2020), trained on the official Cityscapes dataset, which achieves a mIoU of 0.79 on the test set. We utilise its predictions as input to our calibration network, $F_\theta$, which consists of 5 convolutional blocks, each composed of a $3\times3$ convolutional layer, followed by a batch normalisation layer, a leaky ReLU activation, and a dropout layer with 0.1 dropout rate. We pretrain the calibration network in isolation, and subsequently apply it in inference mode while adversarially training the refinement network. We use a batch size of 16, and train with $\mathcal{L}_\text{G}$, estimating $\mathcal{L}_\text{cal}$ with 7 samples from $G_\phi$. The same baseline as in the LIDC experiment is employed, where we replace $\mathcal{L}_\text{cal}$ with $\mathcal{L}_\text{ce}$. As a second control experiment we completely omit the calibration network and instead condition the refinement network on the known ground truth pixelwise categorical distribution over the label. This allows us to directly evaluate the quality of sampling administered from the refinement network.

When training the refinement network with an additional cross entropy loss instead of the calibration loss $\mathcal{L}_\text{cal}$, the predictive distribution collapses, making the output deterministic. Conversely, when we train our refinement network with $\mathcal{L}_\text{cal}$, the learnt predictive distribution is well adjusted, with high diversity and reconstruction quality, significantly outperforming the current state-of-the-art, as shown in Table 2. Fig. 4b displays representative sampled predictions from our model for three input images, and Fig. 4c illustrates the corresponding aleatoric uncertainty maps extracted from $F_\theta(x)$. The learnt multimodality and noise in the dataset is reflected by regions of high uncertainty, where objects belonging to the different stochastic classes consistently display distinct shades of red, corresponding to their respective flip probabilities. Finally, we show that when using the ground truth pixelwise distribution as the input to the refinement network, we attain an almost perfect GED score $(0.038 \pm 0.00)$.

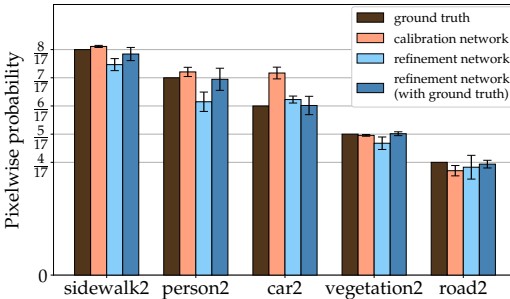

Figure 5: Calibration of the pixelwise probabilities of the five stochastic classes. Note that the calibration network (in orange) is conditioned on black-box predictions.

Table 2: Mean GED scores on the modified Cityscapes. Top section: competing model; middle: $\mathcal{L}_\text{cal}$-regularised cGAN and $\mathcal{L}_\text{ce}$-regularised baseline; bottom: ground truth calibrated cGAN. The GED scores are computed using 16 samples.

| Method | GED |
|---|---|
| Kohl et al. (2018) | $0.206 \pm$ N/A |
| cGAN+$\mathcal{L}_\text{cal}$ | $\mathbf{0.164 \pm 0.01}$ |
| cGAN+$\mathcal{L}_\text{ce}$ | $0.632 \pm 0.07$ |
| cGAN+$\mathcal{L}_\text{cal}$ (gt) | $0.038 \pm 0.00$ |

Since we manually set the flip probabilities for each stochastic class in this dataset, we can directly assess the calibration of our model by comparing the ground truth probabilities to the predicted probabilities from the calibration or refinement network. For $F_\theta$ we use the mean confidence values for each class, and for $G_\phi$ we obtain the empirical class probabilities via $\overline{G_\phi}(F_\theta(x))$, computed from 16 samples (see Appendix A.4 for more details). The ensuing results are shown qualitatively in Fig. 5, which illustrates the calibration of our models on the stochastic classes, evaluated over the entire dataset. This demonstrates that our models are well calibrated, with the calibration offset, computed as the absolute difference between the ground truth and predicted probabilities, being approximately 6% in the worst case (class "car2" for the calibration network). Note that the average calibration offset for $F_\theta(x)$ across the stochastic classes is 1.6%. Further, the ground truth conditioned baseline is almost perfectly calibrated, reflecting the near-optimal GED score reported in Table 2. Thus, we demonstrate that $G_\phi$ learns calibrated refinement of the predictions from $F_\theta$, where the quality of the final predictive distribution depends on the quality of $F_\theta(x)$.

In order to further scrutinise the calibration quality of $F_\theta(x)$, we construct a reliability diagram and compute the corresponding expected calibration error (ECE), as proposed by Guo et al. (2017). To create the diagram each pixel is considered independently and the associated class confidences are binned into 10 equal intervals of size 0.1. We then compute the accuracy for all predictions in each bin. Fig. 6 shows the reliability diagram for the calibration network, where the orange bars depict the calibration gap, defined as the difference between the mean confidence for each interval and the corresponding accuracy. The corresponding ECE score amounts to 2.15%. Note that this also considers the average calibration error computed for the stochastic classes, where we randomly sample the labels according to the defined probabilities. Hence, we confirm that $F_\theta(x)$ is well calibrated.

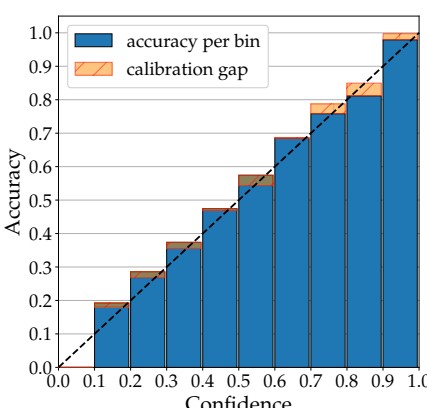

Figure 6: Reliability diagram for the calibration network. ECE = 2.15%.

An important outstanding issue in our approach is that the calibration network may not perfectly capture the class-probabilities, e. g. for the *car2* category in Fig. 5. The limitations of modern neural networks with respect to calibration, as well as possible solutions, are well studied (Guo et al., 2017; Kull et al., 2019; Zhang et al., 2020). Miscalibration has been attributed to several factors, such as long-tailed data distributions, out-of-distribution inputs, specific network architectural elements or optimising procedures. However, since our approach seeks to calibrate relative to the calibration target, and none of the aforementioned issues directly interfere with this process, we consider the problem of absolute calibration to be beyond the scope of this work. Nevertheless we emphasise on its importance for further improving the sample quality.

## 6 CONCLUSION

In this work, we developed a novel framework for semantic segmentation capable of learning a calibrated multimodal predictive distribution, closely matching an estimation of the ground truth distribution of labels. We attained improved results on a modified Cityscapes dataset and competitive scores on the LIDC dataset, indicating the utility of our approach on real-world datasets. We also showed that our approach can be easily integrated into an off-the-shelf, deterministic, black-box semantic segmentation model, enabling sampling an arbitrary number of plausible segmentation maps. By providing multiple valid label proposals and highlighting regions of high data uncertainty, our approach can be used to identify and resolve ambiguities, diminishing risk in safety-critical systems. Therefore, we expect our approach to be particularly beneficial for applications such as map making for autonomous driving or computer-assisted medical diagnostics. Finally, even though the primary focus of this work is semantic segmentation, we demonstrated its versatility through an illustrative toy regression problem, alluding to a broader applicability beyond semantic image segmentation.

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

# Appendices

## A  IMPLEMENTATION DETAILS

In this section we describe the overall training procedure and delve into the training and evaluation details for the stochastic segmentation experiments on the LIDC dataset and the modified Cityscapes dataset.

### A.1  TRAINING PROCEDURE

Algorithm 1 outlines the practical procedure used to pretrain the calibration network, and the subsequent training of the refinement network. Even though the two networks can be trained end-to-end at once, in our experiments we use the two-step training procedure to stabilise the training and reduce the memory consumption on the GPU. This way we are able to fit larger batches and/or more samples for the estimate of $\mathcal{L}_{\text{cal}}$. Algorithm 2 shows the inference procedure for obtaining $M$ output samples.

---

**Algorithm 1** Model training with calibration network pretraining

---

**require:** training data $\mathcal{D}$, number of samples $M$, learning rate $\eta$, calibration loss scale $\lambda$;

1: **procedure** TRAINING($\mathcal{D}, M, \eta, \lambda$)
2:   **while** not converged **do**                                          ▷ Pretraining of $F_\theta$
3:     Sample batch $\{x, y\}_t \in \mathcal{D}$
4:     Update $\theta_{t+1}$ with $-\eta \nabla_{\theta_t} \mathcal{L}_{\text{ce}}(\{x, y\}_t, \theta_t)$
5:   **end while**
6:   **while** not converged **do**                                          ▷ Adversarial training of $G_\phi$ and $D_\psi$
7:     Sample batch $\{x, y\}_t \in \mathcal{D}$
8:     **for** $i = 1, 2, \ldots, M$ **do**
9:       Sample $y_{\text{ref}}^{i,t} = G_{\phi_t}(F_{\theta^*}(x_t), \epsilon_i)$   where $\epsilon_i \sim \mathcal{N}(0, 1)$
10:     **end for**
11:     Compute $\overline{G}_\phi(F_\theta(x))^t = \frac{1}{M} \sum_{i=1}^M y_{\text{ref}}^{i,t}$
12:     Compute $\mathcal{L}_{\text{cal}}(x_t, \theta^*, \phi_t) = \sum_{i,j,k} \left( \overline{G}_\phi(F_\theta(x))^t \left( \log \overline{G}_\phi(F_\theta(x))^t - \log F_{\theta^*}(x_t) \right) \right)_{i,j,k}$
13:     Update $\phi_{t+1}$ with $-\eta \nabla_{\phi_t} \left( \mathcal{L}_{\text{G}}(\theta^*, \phi_t, \{x, y\}_t) + \lambda \mathcal{L}_{\text{cal}}(x_t, \theta^*, \phi_t) \right)$
14:     Update $\psi_{t+1}$ with $-\eta \nabla_{\psi_t} \mathcal{L}_{\text{D}}(\theta^*, \phi_t, \psi_t, \{x, y\}_t)$
15:   **end while**
16: **end procedure**

---

---

**Algorithm 2** Inference procedure

---

**require:** test data point $x$, number of samples $M$;

1: **procedure** INFERENCE($x, M$)                                          ▷ Using $\theta^*$ and $\phi^*$ from Algorithm 1
2:   **for** $i = 1, 2, \ldots, M$ **do**
3:     Sample $y_{\text{ref}}^i = G_{\phi^*}(F_{\theta^*}(x), \epsilon_i)$   where $\epsilon_i \sim \mathcal{N}(0, 1)$
4:   **end for**
5: **end procedure**

---

Notice that any off-the-shelf optimisation algorithm can be used to update the parameters $\theta$, $\phi$ and $\psi$. For the segmentation experiments, we utilise the Adam optimiser (Kingma and Ba, 2014) with $\beta_1 = 0.5$, $\beta_2 = 0.99$ and weight decay of 5e−4. $F_\theta$ is trained with a learning rate of 2e−4 which is then lowered to 1e−4 after 30 epochs. $G_\phi$ and $D_\psi$ are updated according to a schedule, where $G_\phi$ is updated at every iteration, and $D_\psi$ is trained in cycles of 50 iterations of weight updating, followed by 200 iterations with fixed weights. The refinement network is trained with an initial learning rate of 2e−4, lowered to 1e−4 after 30 epochs, whereas the discriminator has an initial learning rate of 1e−5, lowered to 5e−6 after 30 epochs. Additionally, we utilise the $R_1$ zero-centered gradient penalty term (Mescheder et al., 2018), to regularise the discriminator gradient on real data with a weight of 10. Other hyperparameter specifics such as the batch-size and whether we inject stochasticity via random noise samples or latent code samples, depend on the experiment and are disclosed in the respective sections below or in the main text.

## A.2  1D BIMODAL REGRESSION

In the following we derive the mean squared error form of the cross entropy and calibration losses used in experiment Section 5.1 under the assumption that the likelihood model for $q_\theta$ and $q_\phi$ is a univariate Gaussian distribution with a fixed unit variance. Using the setup from Eq. (2) it then follows that:

$$\mathcal{L}_{\text{ce}}(\mathcal{D}, \theta) = -\mathbb{E}_{p_{\mathcal{D}}(x,y)}[\log \mathcal{N}(y \mid F_\theta(x), 1)] \tag{10}$$

$$= \frac{1}{2}\mathbb{E}_{p_{\mathcal{D}}(x,y)}\big[(y - F_\theta(x))^2\big] + \text{const.} \tag{11}$$

Based on the definition of the calibration loss in Eq. (6) we show that:

$$\mathcal{L}_{\text{cal}}(\mathcal{D}, \theta, \phi) = \mathbb{E}_{p_{\mathcal{D}}}\big[\text{KL}\big(\mathcal{N}\big(y \mid \overline{G}_\phi(F_\theta(x)), 1\big) \,\big|\big|\, \mathcal{N}(y \mid F_\theta(x), 1)\big)\big] \tag{12}$$

$$= \mathbb{E}_{p_{\mathcal{D}}}\Big[\mathbb{E}_{\mathcal{N}\big(y|\overline{G}_\phi(F_\theta(x)), 1\big)}\big[\log \mathcal{N}\big(y \mid \overline{G}_\phi(F_\theta(x)), 1\big) - \log \mathcal{N}(y \mid F_\theta(x), 1)\big]\Big] \tag{13}$$

$$= \mathbb{E}_{p_{\mathcal{D}}}\Big[\mathbb{E}_{\mathcal{N}\big(y|\overline{G}_\phi(F_\theta(x)), 1\big)}\Big[-\frac{1}{2}\big(y - \overline{G}_\phi(F_\theta(x))\big)^2 + \frac{1}{2}\big(y - F_\theta(x)\big)^2\Big]\Big] + \text{const} \tag{14}$$

$$= \mathbb{E}_{p_{\mathcal{D}}}\Big[\mathbb{E}_{\mathcal{N}\big(y|\overline{G}_\phi(F_\theta(x)), 1\big)}\Big[y\overline{G}_\phi(F_\theta(x)) - \frac{1}{2}\overline{G}_\phi(F_\theta(x))^2 - yF_\theta(x) + \frac{1}{2}F_\theta(x)^2\Big]\Big] + \text{const} \tag{15}$$

$$= \mathbb{E}_{p_{\mathcal{D}}}\Big[\overline{G}_\phi(F_\theta(x))^2 - \frac{1}{2}\overline{G}_\phi(F_\theta(x))^2 - \overline{G}_\phi(F_\theta(x))F_\theta(x) + \frac{1}{2}F_\theta(x)^2\Big] + \text{const} \tag{16}$$

$$= \frac{1}{2}\mathbb{E}_{p_{\mathcal{D}}}\Big[\big(\overline{G}_\phi(F_\theta(x)) - F_\theta(x)\big)^2\Big] + \text{const.} \tag{17}$$

## A.3  LIDC

**Architectures**   For the calibration network, $F_\theta$, we use the encoder-decoder architecture from SegNet (Badrinarayanan et al., 2017), with a softmax activation on the output layer.

**Training**   During training, we draw random 180×180 image-annotation pairs, and we apply random horizontal flips and crop the data to produce 128×128 lesion-centered image tiles. All of our models were implemented in PyTorch and trained for 80k iterations on a single 32GB Tesla V100 GPU.

We train all our models for the LIDC experiments using 8-dimensional noise vectors in the cGAN experiments, or latent codes in the cVAE-GAN experiments. This value was empirically found to perform well, sufficiently capturing the shape diversity in the dataset. Additionally, in the refinement networks loss, we set the weighting parameter $\lambda$ in the total generator loss, defined in Eq. (7) in the main text, so as to establish a ratio of $\mathcal{L}_{\text{G}} : \mathcal{L}_{\text{cal}} = 1 : 0.5$, where $\mathcal{L}_{\text{G}}$ is the adversarial component of the loss, and $\mathcal{L}_{\text{cal}}$ is the calibration loss component. In practice, the actual weights used are 10 for $\mathcal{L}_{\text{G}}$, and 5 for $\mathcal{L}_{\text{cal}}$.

**Evaluation**   Following Kohl et al. (2018), Kohl et al. (2019), Huang et al. (2018), and Baumgartner et al. (2019) we use the Generalised Energy Distance (GED) (Székely and Rizzo, 2013) metric, given as:

$$D_{\text{GED}}^2(p_{\mathcal{D}}, q_\phi) = 2\mathbb{E}_{s \sim q_\phi, y \sim p_{\mathcal{D}}}[d(s, y)] - \mathbb{E}_{s, s' \sim q_\phi}\big[d(s, s')\big] - \mathbb{E}_{y, y' \sim p_{\mathcal{D}}}\big[d(y, y')\big], \tag{18}$$

where $d(s, y) = 1 - \text{IoU}(s, y)$. Intuitively, the first term of Eq. (18) quantifies the disparity between sampled predictions and the ground truth labels, the second term—the diversity between the predictions, and the third term—the diversity between the ground truth labels. It is important to note that the GED is a sample-based metric, and therefore the quality of the score scales with the number of samples. We approximate the expectations with all 4 ground truth labels ($y \sim p_{\mathcal{D}}$) and 16, 50 or 100 samples from the model ($s \sim q_\phi$) for each input image $x$.

As Kohl et al. (2019) have pointed out, even though the GED metric is a good indicator for how well the learnt predictive distribution fits a multimodal ground truth distribution, it can reward high diversity in sampled predictions even if the individual samples do not show high fidelity to the ground truth distribution. As an alternative metric that is less sensitive to such degenerate cases, Kohl et al. (2019) propose to use the Hungarian-matched IoU (HM-IoU), which finds the optimal 1:1 IoU matching between ground truth samples and sampled predictions. Following Kohl et al. (2019), we duplicate the set of ground truth labels so that the number of both ground truth and predicted samples are 16, and we report the HM-IoU as the average of the best matched pairs for each input image.

In the main text we show the evaluated performance with both GED and HM-IoU metrics over the entire test set and compute the IoU on only the foreground of the sampled labels and predictions. In the case where both

the matched up label and prediction do not show a lesion, the IoU is set to 1, so that a correct prediction of the absence of a lesion is rewarded.

Note that the methods of Baumgartner et al. (2019) and Hu et al. (2019), whose GED scores we report in Table 1, use test sets that differ from the original splits defined in Kohl et al. (2018), which are used in Kohl et al. (2018; 2019) and our work. Baumgartner et al. (2019) uses a random 60:20:20 split for the training, testing and validation sets, and 100 samples to compute the GED score, whereas Hu et al. (2019) use a random 70:15:15 split and 50 samples to compute the GED score. Due to the lack of reproducibility, we do not consider this the conventional way of benchmarking. Therefore, in Table 1 we only report the scores for our models evaluated on the original splits. Nevertheless, we also trained and tested our cGAN+$\mathcal{L}_{cal}$ model on the split methodology defined by Baumgartner et al. (2019) to enable a fairer comparison. This improved our GED score from $0.243 \pm 0.004$ to $0.228 \pm 0.009$, while the HM-IoU, evaluated using 16 samples, remained similar at $0.590 \pm 0.007$ (in the original splits we achieved an HM-IoU score of $0.592 \pm 0.005$). This shows that different random splits can significantly affect the final performance w. r. t. GED score, while HM-IoU appears to be a more robust metric.

## A.4 CITYSCAPES

**Architectures**   For the calibration network $F_\theta$, we design a small neural network with 5 convolutional blocks, each comprised of a 3×3 convolutional layer, followed by a batchnorm layer and a leaky ReLU activation. The network is activated with a softmax function.

**Training**   During training, we apply random horizontal flips, scaling and crops of size 128×128 on the image-label pairs. All of our models were implemented in PyTorch and trained for 120k training iterations on a single 16GB Tesla V100 GPU.

We train all our models for the modified Cityscapes experiments using 32-dimensional noise vectors. Similarly to the LIDC experiments, this value was empirically found to perform well, however, it can be further tuned. As commonly practiced, we use the ignore-masks provided by the Cityscapes dataset to filter out the cross entropy, calibration and adversarial losses during training on the unlabelled pixels. Similarly to our LIDC experiment, we use a weight of 10 for $\mathcal{L}_G$, and 5 for $\mathcal{L}_{cal}$ in the refinement networks loss.

**Evaluation**   The GED metric for Cityscapes is implemented as described in the appendix of Kohl et al. (2018) and evaluated across the entire validation set. In this dataset we have full knowledge of the ground truth class distribution and therefore we compute the GED metric by using the probabilities of each mode directly, as follows:

$$D_{\mathrm{GED}}^2(p_\mathcal{D}, q_\phi) = 2\mathbb{E}_{s \sim q_\phi, y \sim p_\mathcal{D}}[d(s, y)w(y)] - \mathbb{E}_{s, s' \sim q_\phi}[d(s, s')] - \mathbb{E}_{y, y' \sim p_\mathcal{D}}[d(y, y')w(y)w(y')], \quad (19)$$

where $w(\cdot)$ is a function mapping the mode of a given label $y$ to its corresponding probability mass. The distance $d(s, y)$ is computed using the average IoU of the 10 switchable classes only, as done in Kohl et al. (2018). In the cases where none of the switchable classes are present in both the ground truth label and the prediction paired up in $d(s, y)$, the distance score is not considered in the expectation. We use 16 samples to compute the GED score.

For the calibration results presented in Fig. 5, Section 5.2.2 in the main text, we compute the calibration network class-probabilities using the raw predictions of $F_\theta(x)$. We obtain class masks by computing the overlap between the ground truth labels and the black-box predictions for each class. Using these masks we then compute the average class-wise probabilities. The probabilities for the refinement network $G_\phi$ were computed as the average over 16 samples. Here the class masks are obtained by finding the pixels that are specified as the class of interest in the ground truth labels.

# B ADDITIONAL EXPERIMENT RESULTS

To reinforce the results reported in Section 5 we present supplementary results for the bimodal regression experiment and the LIDC and Cityscapes segmentation experiments.

## B.1 1D BIMODAL REGRESSION

Fig. 7 shows the data log-likelihoods for the 9 data configurations for varying mode bias $\pi \in \{0.5, 0.6, 0.9\}$ and mode noise $\sigma \in \{0.01, 0.02, 0.03\}$ trained with and without the calibration loss $\mathcal{L}_{cal}$. Each experiment is repeated 5 times and the individual likelihood curves are plotted in Fig. 7b and Fig. 7d respectively. The results show that high bias is harder to learn, reflected by a slowed down convergence, however, the $\mathcal{L}_{cal}$-regularised model shows greater robustness to weight initialisation. In contrast the non-regularised GAN exhibits mode oscillation expressed as a fluctuation of higher likelihood (one mode is covered) and lower one (between modes).

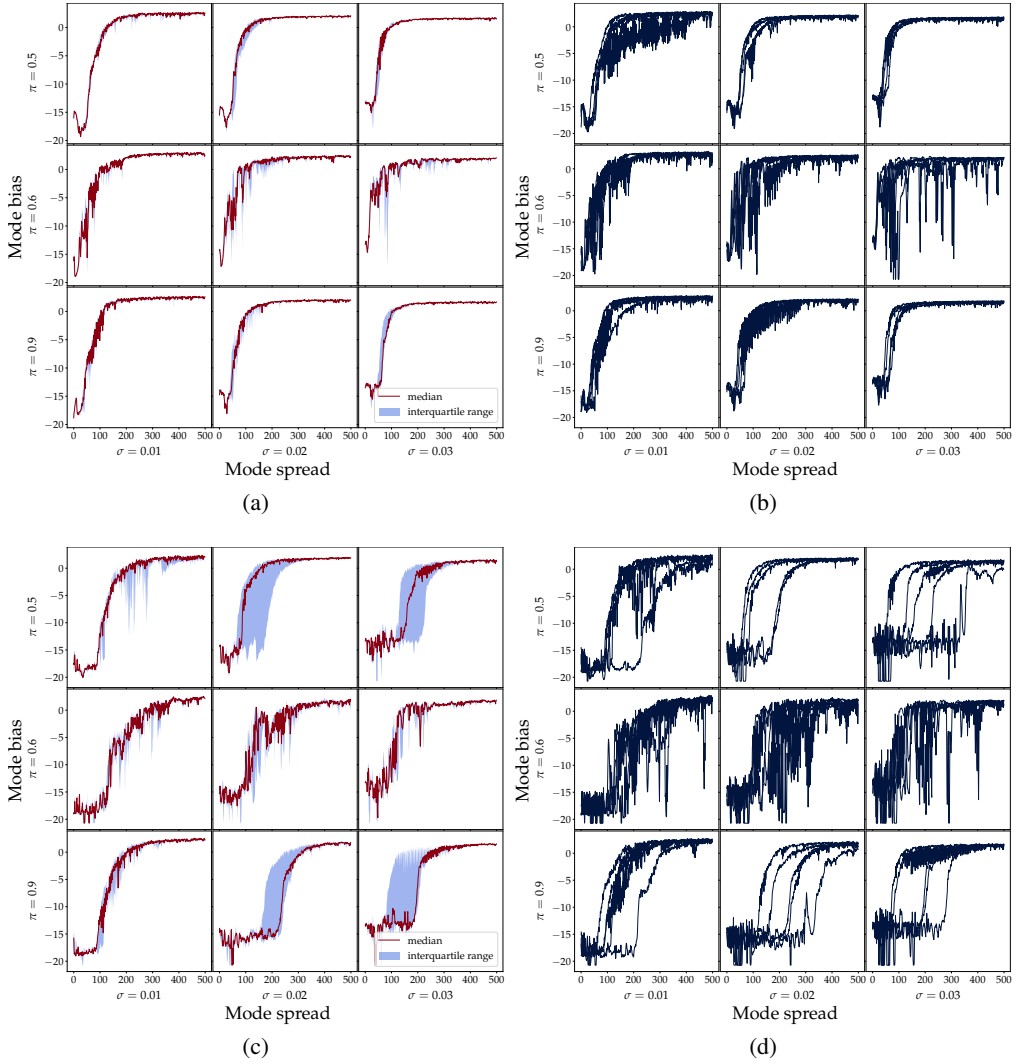

Figure 7: Log-likelihood curves for 5 runs on each of the 9 data configurations. **(a)** No calibration loss ($\lambda = 0$), averaged. **(b)** No calibration loss, individual runs. **(c)** With calibration loss ($\lambda = 1$), averaged. **(d)** With calibration loss, individual runs.

## B.2 LIDC

### B.2.1 QUALITATIVE ANALYSIS

To further examine our $\mathcal{L}_{\text{cal}}$-regularised cGAN model trained on the LIDC dataset, we illustrate representative qualitative results in Fig. 8 and Fig. 9. For every input image $x$, we show the ground truth labels $y_{\text{gt}}^1, \ldots, y_{\text{gt}}^4$ provided by the different expert annotators, overlaying the input image, in the first four columns, and 6 randomly sampled predictions $y_{\text{ref}}^1, \ldots, y_{\text{ref}}^6$ in the last six columns. From left to right, the three columns with the dark blue background in the center of the figures show the average ground truth predictions $\bar{y}_{\text{gt}}$, the output of the calibration network $F_\theta(x)$ and the average of 16 sampled predictions from the refinement network $\bar{y}_{\text{ref}}$. Our results show that even though there is a significant variability between the refinement network samples for a given input image, $\bar{y}_{\text{ref}}$ is almost identical to the calibration target $F_\theta(x)$, due to the diversity regularisation enforced by the calibration loss $\mathcal{L}_{\text{cal}}$.

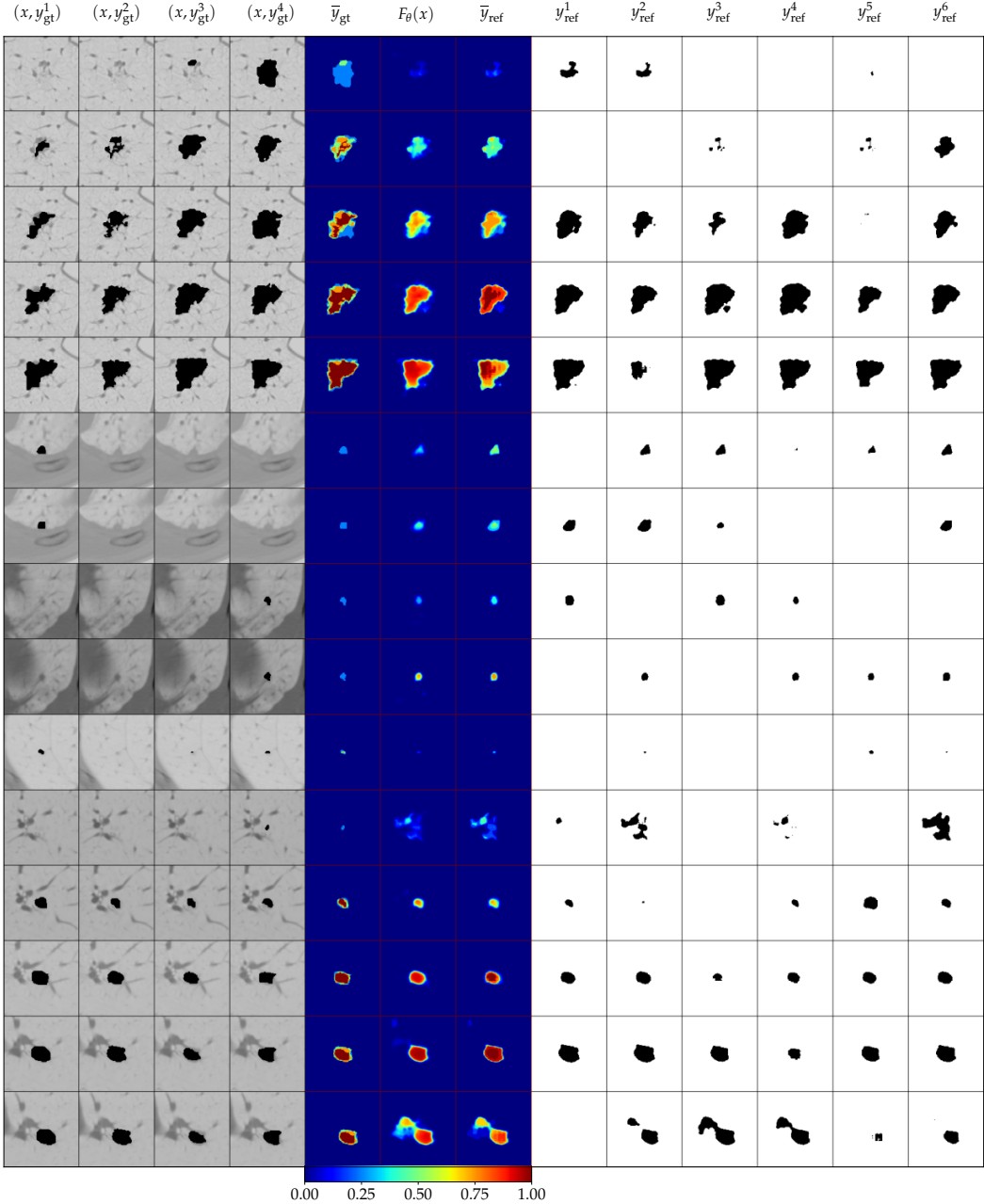

Figure 8: Qualitative results on LIDC samples for the $\mathcal{L}_{\text{cal}}$-regularised cGAN model.

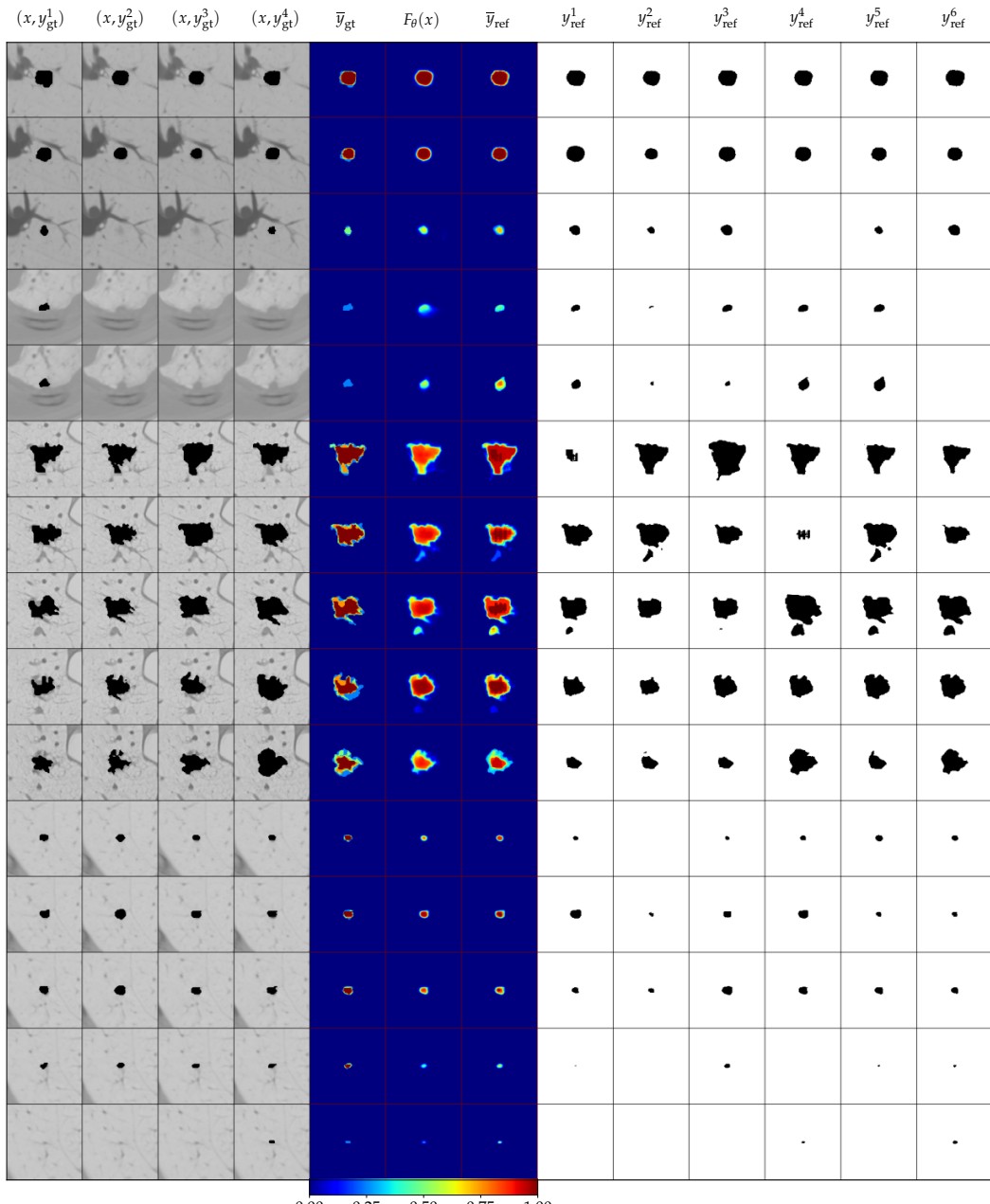

Figure 9: Qualitative results on LIDC samples for the $\mathcal{L}_{\mathrm{cal}}$-regularised cGAN model.

From the qualitative results in Fig. 8 and Fig. 9, it can be seen that the calibration target $F_\theta(x)$ does not always capture well the average of the ground truth distribution $\hat{y}_{\mathrm{gt}}$, affecting the fidelity of the predictive distribution of the refinement network $G_\phi$. This further highlights the importance of future work on improving the calibration of $F_\theta$, e. g. implementing the approaches of Guo et al. (2017); Kull et al. (2019).

### B.2.2 Tuning the number of refinement network samples

Table 3: Mean GED and HM-IoU scores on LIDC for the $\mathcal{L}_{\mathrm{cal}}$-regularised cGAN with 1, 5, 10, 15 and 20 samples. The number of samples used to compute the GED score is denoted in the parentheses in the header of each column. The arrows $\uparrow$ and $\downarrow$ denote if higher or lower score is better.

| Method | GED $\downarrow$ (16) | GED $\downarrow$ (50) | GED $\downarrow$ (100) | HM-IoU $\uparrow$ (16) |
|---|---|---|---|---|
| cGAN+$\mathcal{L}_{\mathrm{cal}}$ (1) | $0.644 \pm 0.033$ | $0.643 \pm 0.033$ | $0.643 \pm 0.033$ | $0.494 \pm 0.013$ |
| cGAN+$\mathcal{L}_{\mathrm{cal}}$ (5) | $0.278 \pm 0.000$ | $0.257 \pm 0.002$ | $0.252 \pm 0.001$ | $0.585 \pm 0.003$ |
| cGAN+$\mathcal{L}_{\mathrm{cal}}$ (10) | $0.277 \pm 0.003$ | $0.257 \pm 0.003$ | $0.250 \pm 0.003$ | $0.589 \pm 0.007$ |
| cGAN+$\mathcal{L}_{\mathrm{cal}}$ (15) | $0.271 \pm 0.002$ | $0.250 \pm 0.001$ | $0.245 \pm 0.003$ | $\mathbf{0.593 \pm 0.002}$ |
| cGAN+$\mathcal{L}_{\mathrm{cal}}$ (20) | $\mathbf{0.264 \pm 0.002}$ | $\mathbf{0.248 \pm 0.004}$ | $\mathbf{0.243 \pm 0.004}$ | $0.592 \pm 0.005$ |

To investigate the effect of the number of samples used to compute $\mathcal{L}_{\mathrm{cal}}$ on the learnt predictive distribution, we experimented on the $\mathcal{L}_{\mathrm{cal}}$-regularised cGAN model using 5, 10, 15 or 20 samples from the refinement network $G_\phi$ during training. As a control experiment, we also train the same model using one sample. Our results, reported in Table 3, show that increasing the number of samples improves the quality of the predictive distribution, whereas using only one sample collapses it. This is expected because increasing the number of samples reduces the variance of the sample mean $\overline{G}_\phi$ and refines the approximation $q_\phi$ of the implicit predictive distribution realised by $G_\phi(x, \epsilon)$. Since in our implementation we reuse the samples from $G_\phi$ in the adversarial component $\mathcal{L}_{\mathrm{G}}$ of the total refinement network loss $\mathcal{L}_{\mathrm{G}}$, the discriminator $D_\psi$ interacts with a larger set of diverse fake samples during each training iteration, thus also improving the quality of $\mathcal{L}_{\mathrm{G}}$.

It is important to note that the benefit of increasing the sample size on the quality of $\mathcal{L}_{\mathrm{cal}}$ highly depends on the intrinsic multimodality in the data. In theory, if the number of samples used matches or exceeds the number of ground truth modes for a given input, it is sufficient to induce a calibrated predictive distribution. However, we usually do not have a priori access to this information. Conversely, if the sample size is too small, the $\mathcal{L}_{\mathrm{cal}}$ loss may introduce bias in the predictive distribution. This could lead to mode coupling or mode collapse, as exemplified in our control experiment with one sample.

In the LIDC dataset, even though we have access to four labels per input image, we argue that the dataset exhibits distributed multimodality, where a given pattern in the input space, e. g. in a patch of pixels, can be associated to many different local labels throughout the dataset. As a result, an input image may correspond to more solutions than the four annotations provided. Therefore increasing the number of samples to more than four shows further improvement in performance. This however can come at the cost of decreased training speed which can be regulated by tuning the sample count parameter while considering the system requirements.

### B.2.3 Inducing multimodality in latent variable models on the LIDC dataset

To examine whether conditioning the source of stochasticity in our model on the input is beneficial, we adapt our framework in order to learn a distribution over a latent code $z$ using variational inference (Graves, 2011). Following most of the existing work on stochastic semantic segmentation (Kohl et al., 2018; 2019; Hu et al., 2019; Baumgartner et al., 2019), we maximise a modified lower bound on the conditional marginal log-likelihood, through a variational distribution $q(z \mid x, y)$. This is realised by minimising the loss function in Higgins et al. (2017), given by:

$$\mathcal{L}_{\mathrm{ELBO}}(x, y) = -\mathbb{E}_{q(z \mid x, y)}[\log p(y \mid x, z)] + \beta \operatorname{KL}(q(z \mid x, y) \,||\, p(z)) \geq -\log p(y \mid x), \qquad (20)$$

where $\beta$ controls the amount of regularisation from a prior distribution $p(z)$ on the approximate posterior $q(z \mid x, y)$. Both $q(z \mid x, y)$ and $p(z)$ are commonly taken as factorised Gaussian distributions.

To this end, we compare our cGAN model to two baselines where the refinement network $G_\phi$ is given as a cVAE-GAN (Larsen et al., 2015). In the first one, we train $G_\phi$ by complementing the adversarial loss $\mathcal{L}_{\mathrm{G}}$ with Eq. (20), using $\beta \in \{0.1, 1, 10\}$ and a fixed standard normal prior. In the second, we introduce a calibration network and train $G_\phi$ using Eq. (7). This does not necessitate specifying a prior.

For a fair comparison, we use the same core models for all of our experiments, introducing only minor modifications to the refinement network to convert the deterministic encoder into a probabilistic one with Gaussian output. This is achieved by splitting the output head of the encoder so as to predict the mean and standard deviation of the encoded distribution (Kingma and Welling, 2013; Kendall and Gal, 2017). Instead of using random noise sampled from a standard Gaussian as our source of stochasticity, the decoder of the refinement network is now injected with latent codes sampled from the Gaussian distribution encoded for each input image. To train the model, we pretrain the $F_\theta$ in isolation, and subsequently apply it in inference mode while training $G_\phi$. We use a batch size of 32, an 8-dimensional latent code, and use 20 samples to compute $\mathcal{L}_{\mathrm{cal}}$.

Table 4: GED and H-IoU scores on LIDC. The top section shows the $\mathcal{L}_{ce}$-regularised baseline and the $\mathcal{L}_{cal}$-regularised cGAN; the bottom section shows baseline and $\mathcal{L}_{cal}$-regularised cVAE-GANs. All $\mathcal{L}_{cal}$-regularised models are trained using 20 samples. The three central columns show the GED score computed with 16, 50 and 100 samples, respectively. The last column shows the HM-IoU score, computed with 16 samples. The arrows ↑ and ↓ indicate whether higher or lower score is better.

| Method | GED ↓ (16) | GED ↓ (50) | GED ↓ (100) | HM-IoU ↑ (16) |
|---|---|---|---|---|
| cGAN+$\mathcal{L}_{ce}$ | $0.639 \pm 0.002$ | — | — | $0.477 \pm 0.004$ |
| cGAN+$\mathcal{L}_{cal}$ | $\mathbf{0.264 \pm 0.002}$ | $\mathbf{0.248 \pm 0.004}$ | $0.243 \pm 0.004$ | $\mathbf{0.592 \pm 0.005}$ |
| cVAE-GAN ($\beta$=0.1) | $0.577 \pm 0.095$ | — | — | $0.484 \pm 0.006$ |
| cVAE-GAN ($\beta$=1) | $0.596 \pm 0.078$ | — | — | $0.474 \pm 0.005$ |
| cVAE-GAN ($\beta$=10) | $0.609 \pm 0.061$ | — | — | $0.482 \pm 0.010$ |
| cVAE-GAN+$\mathcal{L}_{cal}$ ($\beta$=0) | $0.272 \pm 0.006$ | $0.252 \pm 0.006$ | $0.246 \pm 0.006$ | $\mathbf{0.593 \pm 0.003}$ |

We show that the cVAE-GAN model trained with $\mathcal{L}_{cal}$ instead of the traditional complexity loss term, $KL(q(z \mid x, y) \parallel p(z))$ from Eq. (20) is able to learn a distribution over segmentation maps, and performs similarly to our cGAN+$\mathcal{L}_{cal}$ model. This is important because it abrogates the need for specifying a latent-space prior, which is often selected for computational convenience, rather than task relevance (Hafner et al., 2018; Louizos et al., 2019). On the other hand, our cVAE-GAN models trained using the KL-divergence complexity term showed limited diversity, even for large $\beta$ values. The results, shown quantitatively in the bottom part of Table 4, and qualitatively in Fig. 10a, demonstrate that the interaction between $\mathcal{L}_G$ and $\mathcal{L}_{cal}$ can sufficiently induce a multimodal predictive distribution in latent variable models, and indicate that for the purpose of stochastic semantic segmentation the use of a probabilistic encoder is not strictly required.

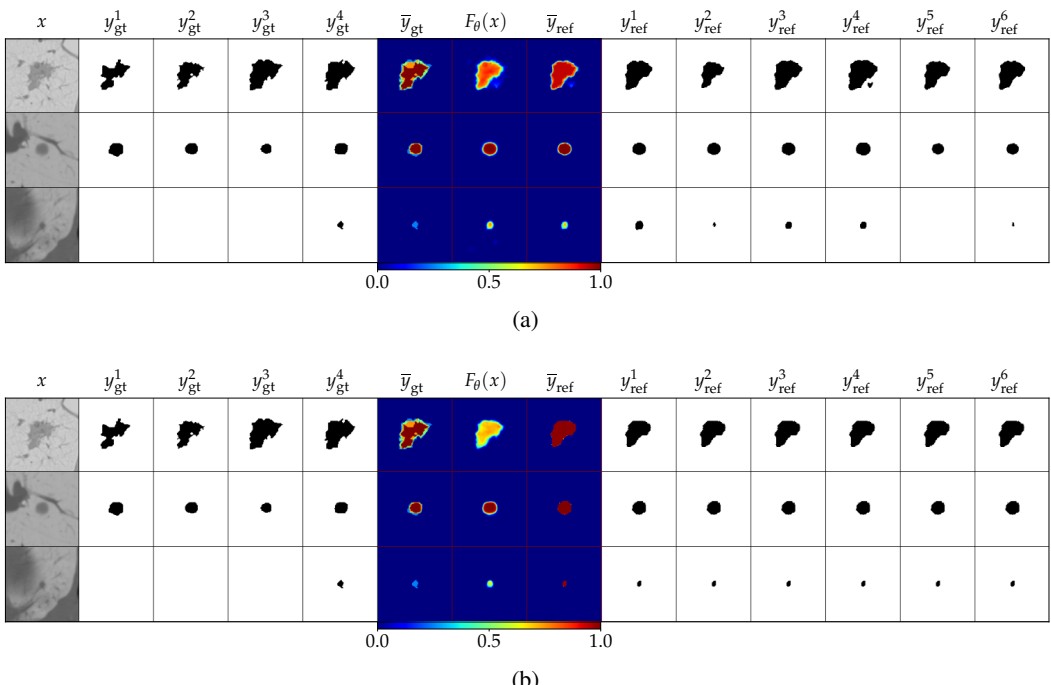

Figure 10: LIDC validation samples for the **(a)** cVAE-GAN and **(b)** cGAN+$\mathcal{L}_{ce}$ baseline model.

## B.3 CITYSCAPES

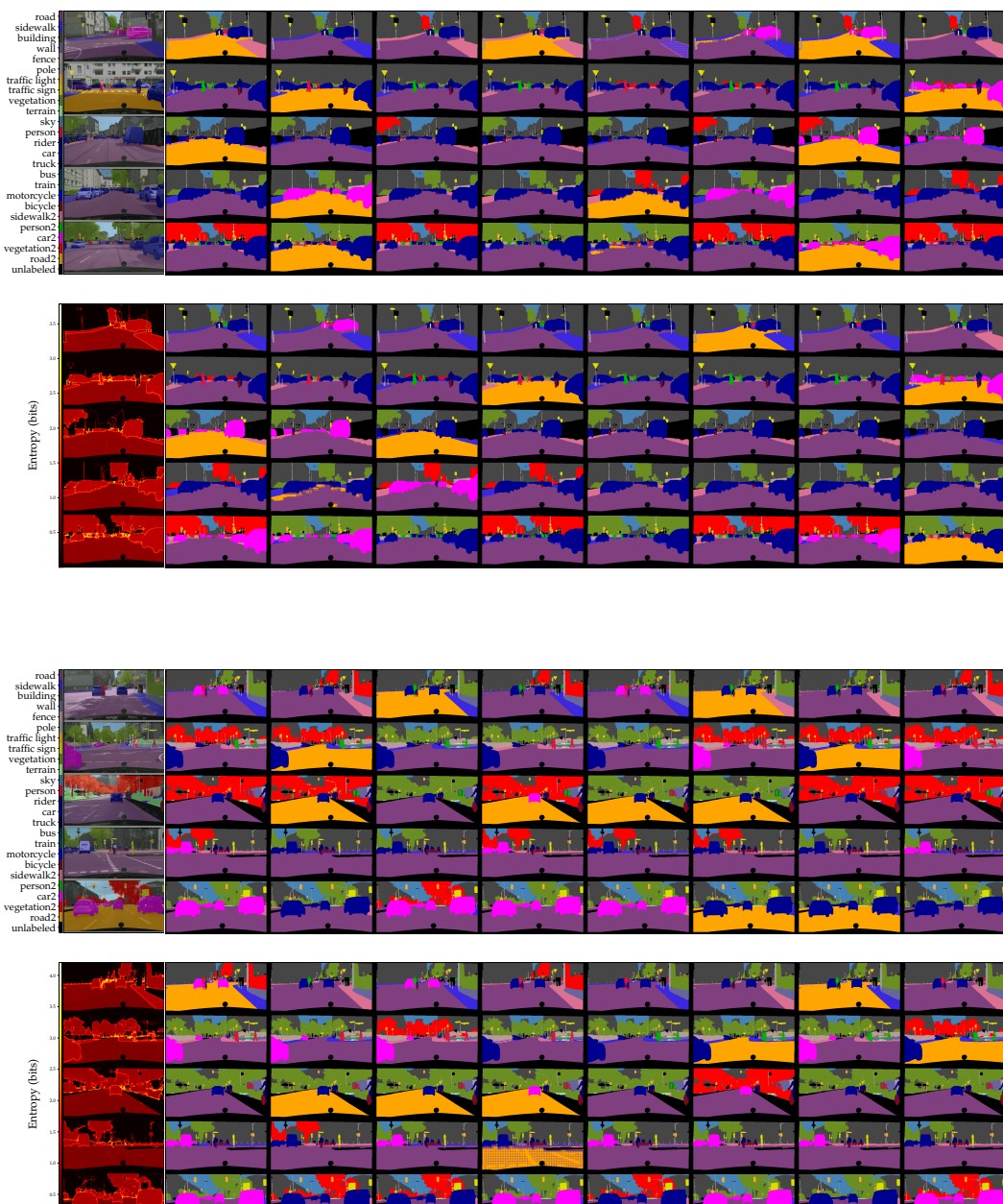

Figure 11: 10 input images, the corresponding aleatoric maps from the calibration network and 16 samples from the refinement network. For visualisation purposes, the samples are split into 8 per row.

### B.3.1 QUALITATIVE ANALYSIS

In this section we provide additional qualitative results for the $\mathcal{L}_{\mathrm{cal}}$-regularised cGAN model trained on the modified Cityscapes dataset (Kohl et al., 2018). In Fig. 11, we show 16 randomly sampled predictions for representative input images $x$, and their corresponding aleatoric uncertainty maps, obtained by computing the entropy of the output of the calibration network, $\mathbb{H}(F_\theta(x))$, as done in Kendall and Gal (2017). The predicted samples are of high quality, evident by object coherence and crisp outlines, and high diversity, where all classes are well represented. Our model effectively learns the entropy of the ground truth distribution in the stochastic classes (*sidewalk*, *person*, *car*, *vegetation* and *road*), as their distinct entropy levels are captured as different

shades of red in the entropy maps, corresponding to the different flip probabilities ($^8/_{17}$, $^7/_{17}$, $^6/_{17}$, $^5/_{17}$ and $^4/_{17}$ respectively). Additionally, it can be seen that edges or object boundaries are also highlighted in the aleatoric uncertainty maps, which reflects inconsistency during manual annotation, which often occurs on input pixels that are difficult to segment.

Fig. 12d shows the entropy of the predictive distribution of the refinement network $G_\phi$, $\mathbb{H}\big(\overline{G}_\phi(F_\theta(x))\big)$, where $\overline{G}_\phi(F_\theta(x))$ is computed as the average of 16 samples from $G_\phi$. Our results demonstrate that $\mathbb{H}\big(\overline{G}_\phi(F_\theta(x))\big)$ is similar to $\mathbb{H}(F_\theta(x))$, depicted in Fig. 12c, as encouraged by the $\mathcal{L}_{\text{cal}}$ regularisation. Notice that object boundaries are also highlighted in $\mathbb{H}\big(\overline{G}_\phi(F_\theta(x))\big)$, indicating that our model captures shape ambiguity as well as class ambiguity. However, some uncertainty information from $\mathbb{H}(F_\theta(x))$ is not present in $\mathbb{H}\big(\overline{G}_\phi(F_\theta(x))\big)$, e. g. the entropy of the different stochastic classes are not always consistent across images, as evident from the different shades of red seen for the road class in Fig. 12d. We expect that increasing the number of samples from the refinement network will improve the aleatoric uncertainty estimates. Nevertheless, the sample-free estimate extracted from $F_\theta(x)$ is cheaper to obtain and more reliable than the sample-based average from $G_\phi$, highlighting an important benefit of our cascaded approach.

Finally, we illustrate in Fig. 12e the high confidence of the predictions from the refinement network $G_\phi(F_\theta(x))$, reflected by their low entropy, $\mathbb{H}(G_\phi(F_\theta(x)))$. This is attributed to the adversarial component in the refinement loss function, which encourages the predictions to assume a one-hot representation, matching the ground truth annotations. Even though each prediction of the refinement network is highly confident, the average of the predictions $\overline{G}_\phi(F_\theta(x))$ is calibrated, as shown in Fig. 12d. This is a clear illustration of the advantage of complementing the adversarial loss term $\mathcal{L}_{\text{G}}$ with the calibration loss term $\mathcal{L}_{\text{cal}}$ in the training objective for the refinement network.

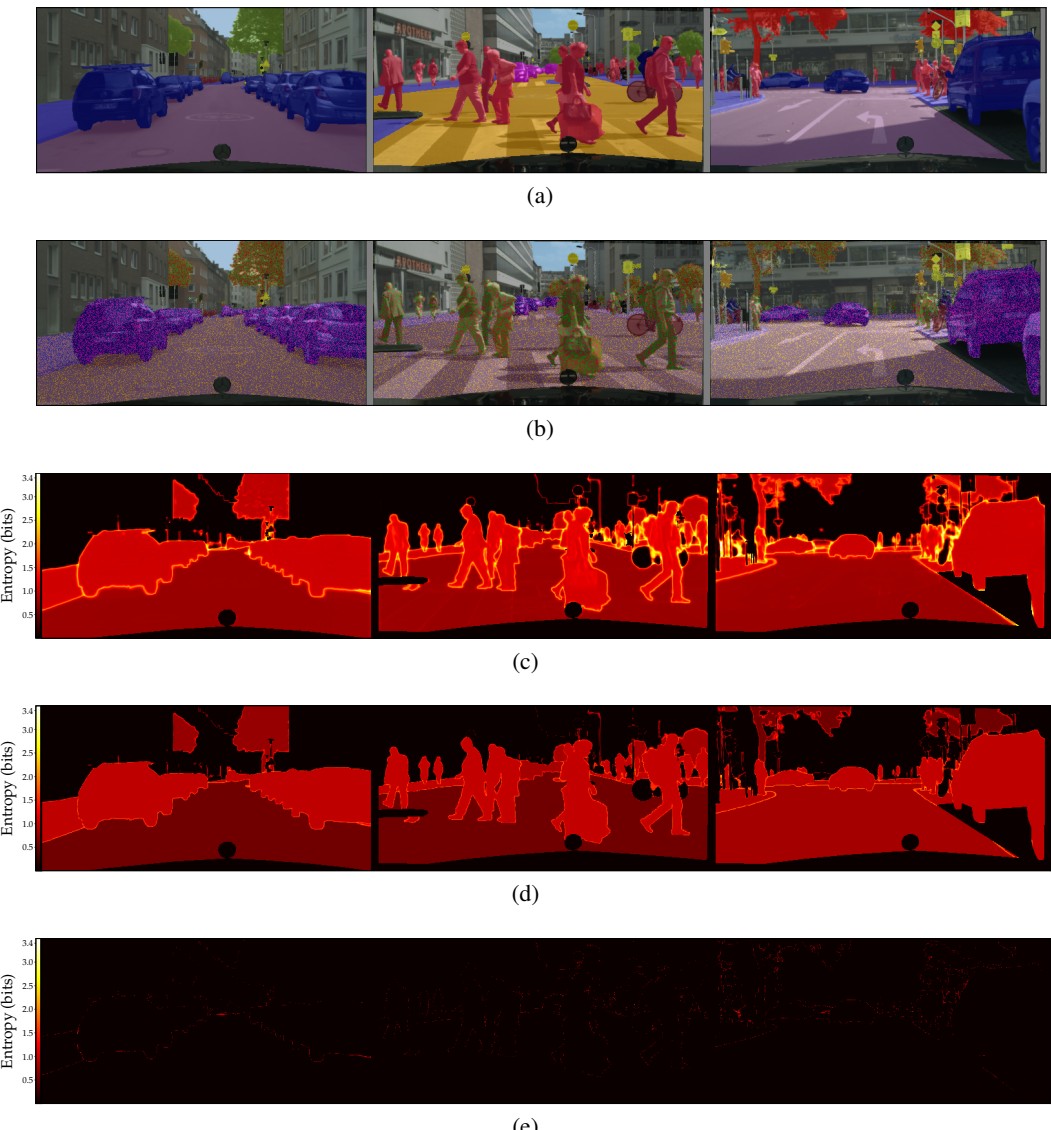

Figure 12: **(a)** Three input images overlaid with the corresponding labels; **(b)** Incoherent samples from the predictive distribution of the calibration network; **(c)** The aleatoric maps from the calibration network; **(d)** Aleatoric maps computed as the entropy of the average of 16 predictions of the refinement network; **(e)** The entropy of one sample of the refinement network output for each input image.

