# OpenReview forum: "Calibrated Adversarial Refinement for Stochastic Semantic Segmentation"
_ICLR.cc/2021/Conference — Reject_

### Official Review · AnonReviewer3 · 2020-10-27
**Interesting idea, but paper and experiments need revision**

**Rating:** 6
**Confidence:** 3

**Review:**

** Summary:
This work addresses the context of semantic segmentation where a single input image could be associated with multiple valid labels, as a result of natural ambiguities. Starting from a pretrained deterministic segmentation network F, this work proposes to use an additional conditional generative model G, named as *refinement network*, to generate multiple segmentation predictions; the model G is conditioned on the segmentation probabilistic output of F and the input image. G is trained with adversarial loss and the proposed *calibration loss*, essentially the KL-divergence between the probabilistic output of F and the sample average of G. At runtime, the unified pipeline of F and G can produce multiple segmentation predictions. On one toy example and two real benchmarks, the proposed method show improvements over addressed baselines, in terms of generalized energy distance (GED) and Hungarian-matched IoU (HM-IoU).

** Strengths:
- The idea of using conditional GANs to produce multiple predictions is interesting.
- The proposed framework and learning scheme are simple. I think it's easy to reimplement and reproduce results.

** Weakness:
- Going through the paper I had trouble understanding how the refinement network G can guarantee to produce calibrated probabilities of segmentation modes. The calibration network F, to my understanding, is a deterministic segmentation model trained in a conventional fashion using only the cross-entropy loss. I believe numerous works proved that a model trained that way will end up with over-confident predictions, which are uncalibrated (actually shown in Figure 5). It actually seems misleading to name F with calibration.

- Outputs of pretrained F is then used to regularize the training of the cGAN G via the KL-divergence "calibration loss" (which is more like a reconstruction loss to me). Can the authors explain how the refinement network, trained to match sample average with uncalibrated probabilistic targets, can successfully produce calibrated probabilistic outcomes? Also, I would love to see results with calibration metrics like NLL and ECE.

- On the Cityscapes experiments, the segmentation network B is finetuned with or without class-flipping labels? It's quite confusing when sometimes F is a full segmentation network as in Sec 3, Sec 4 and Sec 5.2.1, sometimes F is an ad-hoc network like in 5.2.2. Also F's architecture is detailed in the beginning of 5.2 as SegNet, but only used in 5.2.1.

- Can the authors please provide experimental evidence of how the cross-entropy loss and adversarial loss are not well aligned in the presence of noisy data?

- Minor typos:
	+ In Table 2, shouldn't the GED of Kohl et al be 0.206?
	+ It may look obvious but should the notations like H,W,C,K be introduced? I thought C is the number of classes at first.

** Preliminary evaluation: this work targets an interesting task of stochastic semantic segmentation. The architecture design and learning scheme seems reasonable to me. The major problem is the lack of evidence to support the claim on output calibration. In terms of writing, I find the paper hard to follow with lots of confusions. Due to those limitations, I give an initial rating of 5.

-- Post-rebuttal -----------------------------------------------------------------------------------------------

Given the improvement of the last revision, I increase my rating to 6. The revised version has been very much improved, especially in the abstract and introduction Sections. Still I think it's important to additionally have one or two sentences to make very clear on the meaning of calibration, as to not confuse readers.

---

> ### Author Response · Authors · 2020-11-13
> **Response to reviewer #3, Part 1/2**
>
> We would like to thank you for your thorough feedback. We hope to have answered all of your remarks adequately. Below we share our reply to each point separately:
>
> 1. The refinement network is calibrated on the predicted pixel-wise probabilities provided by the calibration network, $F_\theta$, on account of the calibration loss. This works because while the adversarial loss term optimises the refinement network to synthesise confident, label-like predictions (mimicking the one-hot format of the labels), the calibration loss forces the average of multiple such confident predictions to match the probabilities presented by$F_\theta$. Notice that in the central columns in Fig. 3 the average of multiple sampled predictions from the refinement network, $\bar{y}_\text{ref}$, is almost identical to $F_\theta(x)$), on account of the calibration loss. Please let us know if this part is unclear, so that we can update the text accordingly.
> To that end, our basic assumption is that $F_\theta$ can provide accurate estimates of the pixelwise probabilities. It is true that if $F_\theta$ is overconfident, so will be the output of $G$, however, we did not find this to be an issue in our experiments. For example, the predicted probabilities for the stochastic classes in the Cityscapes experiment were well calibrated, as we show in Fig. 5. Regarding your comment that Fig. 5 shows uncalibrated predictions, we argue that on the contrary it illustrates that our predictions are almost perfectly calibrated, as in the worst case (class "car2") the calibration error, which we compute as the absolute difference between the ground truth and predicted probabilities from $F_\theta$ shown in Fig. 5, is approximately 6% (Please note the scale of the y-axis). Furthermore, the average calibration error on the stochastic classes is 1.6%, across three independent runs.
> Finally, we would like to point out that the reliability of the calibration network's predictions can be considered as an orthogonal problem and tackled, among other methods, with the works of Guo et al. (2017); Kull et al. (2019); Zhang et al.(2020). In any case, we show that when the refinement network is conditioned on perfectly accurate pixel-wise probabilities, as explored by our ground truth baseline in the Cityscapes experiment, it is almost perfectly calibrated (see dark brown and dark blue bars in Fig. 5). This demonstrates that the refinement network can indeed perform calibrated adversarial refinement **relative** to the calibration target.
>
> 2. We hope that our reply to your first point has answered your first question here. If you would like us to elaborate further on this, please let us know.
> We did not compute the expected calibration error (ECE), which is the difference in expectation between confidence and accuracy, because in the stochastic Cityscapes experiment we set the flip probabilities for the stochastic classes ourselves. Therefore we can directly visualise the difference between the expected confidence (for the calibration network) or the empirical prediction probabilities (for the refinement network) and their respective ground truth probabilities, as shown in Fig. 5, instead of relying on computing a proxy to the ground truth probabilities (the expected accuracy). To that end, Fig. 5 shows that, even though not absolutely perfect, our predictions at both the calibration network level and the refinement network level are in fact well calibrated. Therefore, as mentioned in our reply to your first point, we didn't find it necessary to apply techniques such as temperature scaling, Dirichlet calibration etc. which could improve the calibration even further. Finally, the approaches we compare to in Tables 1 and 2 have not reported these metrics so we cannot compare to them. Nevertheless, we can add ECE scores if you believe it will help our case.
>
> 3. $B$ is trained on the standard non-stochastic Cityscapes dataset (in the text we mention it is finetuned on the original Cityscapes dataset), but it was never finetuned by us. Rather, the discretised outputs (i.e. in RGB format) from $B$ are fed into the calibration network $F_\theta$, and $F_\theta$ is trained on the stochastic Cityscapes dataset to map $B$'s outputs to the pixel-wise categorical distribution which serves as the calibration target. In this case $B$ can be any off-the-shelf SOTA network or arbitrary non-differentiable segmentation model.
> We apologise for the confusion caused by the change of architecture for the calibration network across experiments. Note that we describe the architecture of the calibration network used for the Cityscapes experiments in the second paragraph of Section 5.2.2. If there are outstanding issues regarding the clarity of the experiment details, please let us know.

---

> > ### Author Response · Authors · 2020-11-13
> > **Response to reviewer #3, Part 2/2**
> >
> > 4. We experimentally demonstrate that combining the cross entropy loss and the adversarial loss on noisy data collapses the predictive distribution of our model to a single, deterministic output (rather than allowing for a multimodal predictive distribution) in the stochastic semantic segmentation experiments through our $\mathcal{L}_\text{ce}$-regularised baseline. A description of this baseline is given at the end of the second paragraph of Section 5.2.1. We show qualitative results in Fig. 9b in the appendix, and quantitative results in Tables 1 and 2.  In practice, downregulating the contribution of the cross entropy loss term can mitigate the problem to some extend, however, the conflict between the two loss terms will still be there. We also gave an intuitive explanation on why this conflict occurs in our response to the 3rd remark by Reviewer 4.
> > We hope that this has sufficiently answered your question. If not, please let us know.
> >
> > 5. Thank you for spotting this, it is indeed 0.206. This is now fixed in the revised version of the paper.
> >
> > 6.  These notations are introduced in the first paragraph in section 3. We will change the text to make it clearer.
> >
> > Last, could you please expand on which parts in the writing were confusing? We would like to improve it as much as we can.

---

> ### Author Response · Authors · 2020-11-20
> **Added ECE score and reliability diagram**
>
> Please note that  we have now added the ECE score (2.15%) and a reliability diagram assessing the calibration network's quality, as suggested in your 2nd point.

---

### Official Review · AnonReviewer1 · 2020-10-27
**Simple yet effective strategy for stochastic segmentation with limitations in the experimental evaluation**

**Rating:** 6
**Confidence:** 3

**Review:**

Summary:
This paper presents an approach to stochastic semantic segmentation. The proposed strategy involves a simple extension of neural network architectures for semantic segmentation. In particular, the probabilistic output of a semantic segmentation network is fed into a GAN, which generates a final segmentation. In addition to the classic loss, the GAN is trained such that the average of its prediction matches the input distribution, hence calibrating the distribution. The experiments on a toy dataset and two segmentation datasets demonstrate the superiority of the proposed approach compared to using standard segmentation loss and a few related works.

Pros:
- The writing is good, and the method is explained intuitively
- The proposed approach is simple yet seems effective
- The proposed approach is modular and can be applied to pre-trained network architectures

Cons:
- My major concern is that the experimental evaluation does not sufficiently demonstrate that the proposed module is indeed necessary. In the preliminaries section, the authors discuss several simpler alternatives that are not tested in the experiments. E.g., that direct sampling from q_theta yields incoherent segmentation maps or that combining the generator loss with the pixel-wise loss in Eq.2 is not sufficient. I think these settings would serve as good additional baselines in the experimental evaluation.
- A minor additional criticism is that the authors do not compare to other relevant work such as Kendall and Gal 2017. Moreover, on the Cityscapes dataset, they only compare to one single baseline.

--------------------------------------------------------------------------------------------------------------------------
Post rebuttal

I thank the authors for providing detailed answers to my concerns. Considering the concerns of the other reviewers and the authors' answers and additional experiments, I think that the paper provides a sufficient contribution to an important research topic. Therefore, I retain my initial rating.

---

> ### Author Response · Authors · 2020-11-13
> **Response to reviewer #1**
>
> Thank you very much for your feedback. We provide clarifications to your remarks below.
>
> 1. As we have discussed in the second paragraph of Section 3, direct sampling from $q_{\theta}$ yields incoherent segmentation maps in regions of inter-label inconsistencies (regions where the output space is noisy), because the calibration network is optimised with the pixelwise loss from (Eq. 2), and thereby all pixels are modelled independently. For example, in a binary segmentation task, let [0.5, 0.5] be the probability assigned to each of two adjacent pixels belonging to the same semantic object. Then direct sampling could lead to one pixel belonging to class 1, and the other pixel belonging to class 2, while a self-consistent solution requires that both belong to the same class. Therefore, for our use case we did not consider this as a suitable baseline. To further illustrate this, we added an example in Fig. 11b, Appendix B.3.1.
> Regarding your comment on combining the generator loss (we assume you mean the adversarial loss here) with the pixel-wise loss in Eq. 2, we have already used this as a baseline in our stochastic semantic segmentation experiments, and refer to it as the cGAN+$\mathcal{L}_\text{ce}$ baseline. As evident from the corresponding results in Tables 1 and 2, and Fig. 9b in the appendix, this leads to a collapsed predictive distribution, in support of our claim that combining the adversarial and cross entropy losses is not sufficient to express multimodality. If you have any further questions regarding these points, please let us know.
>
> 2. We agree that a comparison of our model to that proposed by Kendall and Gal (2017) would be insightful for assessing how well our calibration network captures uncertainty in multimodal datasets, and such an experiment could constitute important, but nonetheless orthogonal, future work. However, we did not perform this comparison here because we focused on evaluating the quality of the sampled predictions obtained by the refinement network; that is, how well the predictive distribution of the refinement network fits a multimodal ground truth distribution. Note that the model proposed by Kendall and Gal (2017) is specifically designed to model uncertainty, but it is not used to predict diverse, self-consistent segmentation maps, which is why we did not consider it as baseline when assessing the quality of the refinement network's predictions.
> Regarding the stochastic Cityscapes experiment, to the best of our knowledge, only the original authors, Kohl et al. (2018), have experimented on this version of the dataset beside us. We would have liked to have more baselines there to make our case stronger.

---

### Official Review · AnonReviewer2 · 2020-10-28
**Interesting method but with missing comparisons and metrics**

**Rating:** 6
**Confidence:** 4

**Review:**

-----------------------------------------------------------------------------------------------------------------------------------------------------------------
POST REBUTTAL
-----------------------------------------------------------------------------------------------------------------------------------------------------------------

The rebuttal has addressed most of my concerns and I am happy to increase the score.

-----------------------------------------------------------------------------------------------------------------------------------------------------------------

The main strengths of the work are -
* The approach is relatively novel and addresses an important problem.
* The paper clearly highlights issues with prior work e.g. generator loss is often complemented with pixelwise loss, however, these two objective functions are not well aligned in the presence of noisy data.
* The paper includes experiments on toy data -- which highlight the contributions of the paper.
* The proposed approach outperforms prior work -- Hu et al. (2019), Baumgartner et al. (2019) on the LIDC dataset.
* The proposed approach outperforms Kohl et al. (2018) on CityScapes in case of the GED metric.

The main weaknesses are,
* No comparison with closely related approach [1] -- which also proposes a Bayesian approach to capture a calibrated multimodal predictive distributions. In fact, the experiments on 1d bimodal toy data are similar to the experiment in Figure 1 in [1]. A detailed comparison is necessary.
* Experiments on 1D bimodal data -- the baseline $G_\phi(F_\theta(x), \epsilon)$ is a typical conditional GAN? This seems to be a weak baseline, as recent works [3] address the model collapse issues of conditional GANs. Strong baselines e.g. [1,3], conditional VAEs etc should be considered.
* Experiments on CityScapes -- the paper does not show results using metrics used by prior work [2] -- in particular Precision-Recall curves and calibration plots which shows the frequency of correctly predicted labels for each bin of predicted probability values. These metrics are also used in [1]. These metrics would better illustrate the calibration of predictions of the proposed approach.
* Several unclarities -- What is the contribution of the two components - Calibration network and Refinement network on the calibration of the final output? Does the Refinement network aid in improving calibration?

[1] Bayesian Prediction of Future Street Scenes using Synthetic Likelihoods, ICLR 2019.
[2] What Uncertainties Do We Need in Bayesian Deep Learning for Computer Vision?, NeurIPS 2017.
[3] Diversity-Sensitive Conditional Generative Adversarial Networks, ICLR 2019.

---

> ### Author Response · Authors · 2020-11-15
> **Response to reviewer #2**
>
> Thank you very much for your remarks. We have changed the manuscript to address these, and answered your points one by one below.
>
> 1. Thank you very much for referring us to this related work. Unfortunately we were not aware of it. We have now added a paragraph in Section 2, where we discuss the proposed method of [1] and the differences to our approach.
>
> 2. In the 1D bimodal regression task, we indeed compare our model cGAN+$\mathcal{L}_\text{cal}$ with a typical cGAN (baseline), conditioned on the output of the calibration network. The aim of this experiment was to single out the contribution of the calibration loss in a visually intuitive way, rather than proving that our method is better than other competing methods (we instead use the stochastic segmentation experiments for this purpose). Therefore we deemed using other baselines here unnecessary. We acknowledge that the works of [1] and [3] would probably also prevent mode collapse in this case but neither explicitly addresses the calibration of the predictive distribution.
> Further we use this experiment to demonstrate that our approach is more general than semantic image segmentation, and can be readily adapted to regression tasks too.
> Note that we now also added a reference to [3] in our related work section.
>
> 3. We do not show precision-recall curves in our stochastic semantic segmentation experiments because similarly to the IoU metric, these are not well suited for multimodal distributions as they are designed to compare a deterministic prediction with a unique ground truth label, for any given data point. However, in the present case, we purposely consider cases where there are multiple ground truth labels for a given input. Therefore, situations can arise where perfectly valid predictions are unduly penalised because they are matched with ground truth labels from a different valid mode. Further, the expected IoU over multiple prediction-label pairs is not sensitive to diversity within sampled predictions or within ground truth labels, which is why the GED score has two additional terms that compute the prediction diversity as the expected IoU between sampled predictions, and label diversity as the expected IoU between ground truth labels. We provide an intuitive explanation for GED in the "Evaluation" paragraphs of Appendices A.3 and A.4. Furthermore, using the GED and HM-IoU metrics allows us to compare to related works, which also use these metrics only.
> Regarding your comment on calibration plots, we do not compute the calibration plots as accuracy vs. confidence because we already have access to the ground truth probabilities for the stochastic classes. Therefore we can directly compare the refinement network's empirical probabilities, obtained through averaging ($\overline{G}_{\phi}(F_\theta(x))$), with the mentioned ground truth probabilities, which is what we show in Fig. 5. Please let us know if this answer is not sufficient.
>
> 4. In the current setup, the refinement network is calibrated relative to the calibration target set by the calibration network. However, the refinement network can learn to ignore erroneous pixel probabilities in the calibration target if they are not realistic, on account of the adversarial loss (e.g. can ignore heterogeneity in probabilities for pixels belonging to the same object, in favour of self-consistent predictions). Therefore, while the calibration network is the main driver to overall calibration in our model, the refinement network can indeed also implicitly affect the calibration of the final predictive distribution in order to satisfy the adversarial objective while minimising the calibration loss. If you find this answer insightful, we can add it in the manuscript to resolve any unclarities.

---

> ### Author Response · Authors · 2020-11-20
> **Added ECE score and reliability diagram**
>
> Please note that we have now added the ECE score (2.15%) and a reliability diagram assessing the calibration network's quality, addressing your 3rd point.

---

### Official Review · AnonReviewer4 · 2020-10-29
**The authors propose a two-stage calibration technique for semantic segmentation.  The approach is conceptually simple and can be applied to a number of base segmentation models.  However, there is a lack of strong justification for their approach, seemingly poor pixel-wise calibration results, and the main novel contribution is a simple additive KL Divergence term in the generator's objective.**

**Rating:** 4
**Confidence:** 4

**Review:**

In this work the authors propose a two-stage, adversarial training technique to calibrate a semantic segmentation model when faced with conflicting labels in the train set.  Their approach is to first train a segmentation model. Then, it is used to feed a GAN model which itself is trained against a discriminator to produce diverse segmentations that reflect the diversity in the train set.  The authors compare their approach to similar methods on a synthetic data set and two semantic segmentation data sets.

Pros:
1) The technique is conceptually simple as training a segmentation model with cross entropy loss as well as training a GAN are both well-understood.
2) The technique can really be used to calibrate any differentiable semantic segmentation model, regardless of who trained it or how.
3) The experimental details are described in enough detail to likely be able to replicate most of the results
4) Minus some organizational issues, the writing is good overall.


Cons:
1) I found the motivation in the introduction to be slightly difficult to follow.  More specifically, the concepts ambiguity in data labels leading to a multi-modal data distribution and the need to model uncertainty need to be more tightly discussed.  As it is written now, the authors first argue for modeling a noisy empirical distribution that captures the ambiguities, which seems counter intuitive since there assumedly exists a single true segmentation of the image then later discuss uncertainty.  I think discussing uncertainty modeling, and specifically calibration, first would alleviate this issue.
2) "Isola et al. (2017) have demonstrated that it introduces only minor stochasticity in the output and returns inconsistent samples." (page 2) - In Isola et al. (2017)  dropout is used for image to image translation in a GAN.  I do not think this provides sufficient evidence that using dropout BNNs  such as in Kendal and Gal (2017) has these properties.
3) The authors argue that combining the GAN loss and the pixel-wise cross entropy is not well-suited for noisy data because they are often at odds.  It's not clear to me that this is a problem in practice, as the cross entropy will spread probability mass across different conflicting labeled instances.  Assume a pixel in the data set appears twice with two different labels.  The sum of the cross entropy over these two examples is minimized by a model that puts probability 0.5 on each of the two classes.  This seems like the multi-modal behavior the authors argue for.  Perhaps there is something more subtle going on, but the lack of formal analysis makes it difficult to understand how much of an issue it is.  Further, the proposed method is adding a KL Divergence term to the generator loss that is at odds with the GAN loss, which was what the authors argue against.  In short, the argument against the most similar method and the proposed method is weak in my opinion.
4) I think the name calibration network is a bit misleading. It seems the calibration network is the base segmentation model and the refinement network is calibrating the model.
5) The paper would benefit from an algorithm sketch to explicitly show the two stages of training.
6) In the experiments the authors switch to mean squared loss after arguing against cross-entropy.  It's not clear why this is done or why this is a proper baseline.
7) Looking at figure 3, it would seem the refinement network does not generate outputs that closely match any of the ground truth annotations, but rather some combinations of them.  In practice, I would assume that someone would use the mean and standard deviation to understand the predictions and not samples, so this is less of an issue, but it highlights an issue with considering each pixel independently.
8) While I think the GED metric make sense here, I think calibration (like expected calibration error) or uncertainty focused (like those proposed in (Mukhoti and Gal; 2019)) would be useful to tell a more complete story of the evaluation.
9) Figure 5 is unclear.  The text seems to imply that this shows that their technique does not calibrate their model well.  To me this is a strong argument against their approach.  I am not sure the value of a diverse set of segmentations if the model cannot accurately convey uncertainty in predictions.

---

> ### Author Response · Authors · 2020-11-12
> **Response to reviewer #4, Part 1/2**
>
> Thank you very much for you detailed feedback. We have addressed all of your points and updated the manuscript accordingly. Below we share our reply to each point separately:
>
> 1. We believe there is a misunderstanding. We specifically study cases with multiple valid segmentation labels for a single input image. For example, in the LIDC dataset all 4 expert annotations are considered as ground truth. This establishes measurable aleatoric uncertainty in the label space, which we model with the calibration network, e.g. see the middle part of Fig. 3 or Fig. 4c. We explicitly mention this in the abstract, and refer to this property (having multiple valid interpretations) in the introduction using the word "ambiguous". Does this clarify your concern?
>
> 2. This is not our own conclusion, but that of the cited authors (see last paragraph of section 3.1 in Isola et al., (2017)). This is also supported by Kohl et al. (2018), with their MC-dropout baseline. Further, Kendal and Gal (2017) used dropout Bayesian NNs to extract epistemic uncertainty estimates but not to produce diverse and self-consistent predictions.
>
> 3. Taking your example into consideration, we agree that the calibration network will learn to assign [0.5, 0.5] probability to each outcome (class1 or class2). However a discriminator will have an easy task in identifying this as a fake example because a ground truth label is either of class1, i.e. [1, 0] or class2, [0, 1], but never both. In essence, on noisy data points cross entropy optimisation leads to a single mode-averaging solution, and therefore pushes for unconfident predictions, whereas adversarial optimisation allows for multiple solutions but pushes for confident predictions. This "conflicting" signal to the generator is what we describe in the last paragraph of Section 3 in the paper. We show that this leads to a collapsed predictive distribution, as evident from the results for the cGAN+$\mathcal{L}_\text{ce}$ baselines in Tables 1 and 2, and Fig. 9b in the appendix.
> Regarding KL-divergence term, in contrast to cross entropy loss, the calibration loss term does not constrain the model to a single solution. This is because it is expressed as a KL-divergence between the *average* of multiple predictions (e.g. mean of {[1, 0], [0, 1]}) and the calibration target ([0.5, 0.5]). This is elaborated in Section 4.1, page 4 of the main document, right before we define Equation 7. If this does not answer your concern, please let us know.
>
> 4. We call the calibration network as such because it predicts a calibration target for the refinement network, which is a pixelwise categorical distribution over the semantic classes. The refinement network's predictive distribution is then calibrated according to this calibration target via the calibration loss. Perhaps there is indeed a better name for the network in question, and we are open to changing it if an appropriate alternative is suggested.
>
> 5. We have included an algorithmic sketch in the supplementary material in section A.1 but not in the main text, due to the strict page limit for the submission. We now added a note in the practical considerations paragraph (Section 4.2, page 4) referring to the respective figure in the appendix. If it is important as per the reader's perspective, and space permits it we will move it in the main text.
>
> 6. In the experiment in question (1D toy regression task) we show in a visually intuitive way that the proposed calibration mechanism is not limited to categorical distributions only. To that end, we take the assumption that the data is Gaussian-distributed with a fixed variance, which reduces both the cross entropy and calibration losses to mean squared error, as described in the second paragraph of Section 5.1., page 5. We added a derivation of this result under Appendix A.2. Note that all general remarks about the differences between the calibration and cross entropy losses also hold for this regression experiment. We hope to have explained it clearly now. Please let us know in case you would like further explanation.

---

> > ### Author Response · Authors · 2020-11-12
> > **Response to reviewer #4, Part 2/2**
> >
> > 7. We believe that there is some confusion arising from the fact that the samples shown in Fig. 3 are not ordered to match annotators 1 to 4 but rather presented as an unordered set of 6 random samples. We actually show quantitatively in Table 1, using both the Hungarian-matched IoU (HM-IoU) and GED metrics, that our samples correspond to the 4 ground-truth labels better than other SOTA methods. We show more qualitative results in Fig. 7 and 8 in Appendix B.2.1, illustrating that in general our predictions capture the salient features in individual ground truth labels. Finally, rather than showcasing the fidelity of our best samples, we use Fig. 3 to show that even though the average of the sampled predictions from the refinement network, $\overline{G}_{\phi}(F_\theta(x))$, is almost identical to the calibration target, $F_\theta(x)$, the individual sampled predictions show high diversity.
> > Regarding your comment "but it highlights an issue with considering each pixel independently", could you elaborate further what you mean?
> >
> > 8. In this work, we focus on assessing the quality (fidelity and diversity) of the sampled predictions of the refinement network using the GED and HM-IoU metrics, which is sufficient for the presented use cases. In terms of the calibration, we indicate qualitative results on the calibration of our model on the stochastic classes in Fig. 5. Finally, we see the assessment of our uncertainty estimates as well as the improvement of the calibration network as important future work complementing the method presented in this paper.
> >
> > 9. We are confused as to why you would conclude this, because we show in Fig. 5 that we **can** calibrate. To support this fact, note that the largest error in pixelwise class probability is about 6\% (note the scale of the y-axis), which we also discussed in Appendix B.3.1. We can move the discussion to the main text, given the increased page limit. If this does not clarify it, could you elaborate on what part is unclear?

---

> > > ### Comment · AnonReviewer4 · 2020-11-12
> > > **Comments on Responses**
> > >
> > > Thank you for responding to my initial feedback.  I'll try to address the most important comments, and I have. updated my review score to reflect my better understanding of the work.
> > >
> > > 1. I believe I understand the claims.  My issue is with the way it is presented.  I think the work would be better served by making a more clear connection made between the tasks of obtaining a well calibrated model (for some formal definition of “well-calibrated’) and generating a diverse set of predictions.  To me, it is not clear that the latter implies the former, and the way I read the introduction it seems like that is the claim.  Conversely, Mukhoti and Gal; 2019 show their technique calibrates a segmentation model well according to their proposed metrics, but this paper (the one I am reviewing here) claims this does not provide a diverse set of predictions.  Why is it important that it does?  What are use cases where someone needs to view diverse segmentations instead of uncertainty estimates? How do these segmentations directly affect calibration or is it a secondary effect?
> > >
> > > 2. My point is that in the Isola et al. paper, they add dropout to a GAN, which is both adversarially trained and is a function of a noise vector.  MC-Dropout as it was originally proposed did not have either of these properties and has since been applied to segmentation in Mukhoti and Gal; 2019. Because of this I think it is difficult to state definitively how diverse samples from such a segmentation model would be.  Honestly, this is a fairly nuanced and possibly not a vital distinction to make in the paper.
> > >
> > > 3. Thank you for clarifying.  I see how the conflict between the two terms conflict in the proposed object. And how the two stage training (e.g. training for each term in separate steps) relieves this.  It’s still not clear to me why the KL divergence term is necessary.  It would seem the GAN loss w.r.t the refinement network parameters is minimized when the refinement network outputs each of the conflicting ground truth segmentations, which would make the empirical mean of those equal to the empirical mean of the conflicting segmentations.  As I discussed in my previous comment, that is what the cross-entropy in the calibration network should do.  In short, it’s not clear to me why the GAN loss doesn’t already minimize the KL divergence between the two network outputs.
> > >
> > > 6. This helps my understanding thank you.  I think this experiment might detract from the paper, as it poses a slightly different learning problem than the main focus, and is a very simple synthetic experiment.
> > >
> > > 7. Just to be clear, the four segmentations immediately to the right of the input image are from the four expert annotators and the six segmentations all the way to the right were drawn from the refinement network, correct?  This was my understanding.  While critiquing qualitative results in any constructive way is difficult, I think this is illustrative of how practitioner would use the proposed model (visualizing a heat map map of where the model is uncertain in its segmentation).  Because of that, quantitatively comparing the probabilities expressed in the heat maps may be appropriate here.  GED is close, but does not quite articulate it.
> > >
> > > 8. I think the confusion is in that calibration mentioned throughout the paper and the title.  As such, one could reasonably expect calibration-centric metrics.
> > >
> > > 9. I do not think I understand Figure 5 at all, then.  I think there needs to be much more elaboration. I feel the caption and the body of the text is insufficient for the reader to fully understand what is being conveyed.  Specifically, there needs to be some discussion of how to interpret the graph.  Typical calibration metrics are expressed as some kind of error between the predicted probabilities and the ground truth.  Can this graph be interpreted as such?  I’m struggling to get an idea of that with the graph and what little discussion there is about it.

---

> > > > ### Author Response · Authors · 2020-11-16
> > > > **Replies to comments on responses.**
> > > >
> > > > 1. In that case we apologise for misunderstanding your question. Indeed we treat the quality of being well calibrated, and the ability of generating a diverse set of predictions as orthogonal notions. Regarding use-cases, we expect our model to be particularly useful for hypothesis-driven reasoning in human-in-the-loop semi-automatic settings. For example, large scale manual annotation of segmentation maps is very labour-intensive (each label in Cityscapes takes on average around 1.5 hours to annotate (Cordts et al. (2016))). It would be cost-efficient for the annotators to be able to chose among plausible predictions rather than to redraw from scratch. Ideally the diversity in the proposals should correspond to the uncertainty in the data. We have now added a similar paragraph to this one in the introduction to help our motivation for designing this approach.
> > > >
> > > > 2. Thank you for the explanation, we have indeed misunderstood your first remark. We have now updated the PDF, adding the mentioned work regarding the uncertainty estimate. But indeed, it does not change the theory presented in the paper nor its experimental validation.
> > > >
> > > > 3. Thank you for elaborating on your original question. Actually, that may not always work.  It is well documented in literature that cGANs tend to mode collapse, unless sufficiently regularised. This occurs because the discriminator is satisfied as long as the input segmentation map is realistic, therefore the generator (our refinement network) is not incentivised to match noise vectors to different ground truth modes. Such a regularisation is imposed by our calibration loss, as illustrated in Fig. 2. Please let us know if you would like further clarification.
> > > >
> > > > 4. Note: Point 4 here addresses the 6th point in the initial review.
> > > > We believe that it is useful to visualise how the calibration loss allows us to learn a calibrated multimodal predictive distribution, and how it can easily be translated to regression tasks, showcasing the flexibility of our method. Additionally, Fig. 2 provides illustrative evidence supporting our reply to your previous point above, demonstrating that the refinement network without our proposed calibration loss can learn to sample only a single mode (see Fig. 2c), however, the calibration loss forces the refinement network to learn to output each of the conflicting ground truth data points in the correct proportion, as shown in Fig. 2b. We can make this clearer in the main text (Section 5.1) if you think that it will help the motivation for this experiment.
> > > >
> > > > 5. Note: Point 5 here addresses the 7th point in the initial review.
> > > > Yes indeed, you understand the figure correctly.
> > > > Regarding quantitative evaluation, we could compute the differences between the probabilities in the heat maps provided, however, the related methods we consider as baselines do not provide such estimates, and therefore we cannot compare to them. We agree with you that GED has its shortcomings, which is why we use the HM-IoU metric, introduced by Kohl et. al (2019). This was already explained in the paper at time of submission (under Appendix A.3) and we now moved it to the Metrics paragraph in Section 5.2, to provide a better explanation in the main text and prevent further confusion.
> > > >
> > > > 6. Note: Point 6 here addresses the 8th point in the initial review.
> > > > This is a fair point indeed. We have updated the introduction and the conclusion to make it explicit that we calibrate relative to the calibration target provided by the first stage model, which may or may not be absolutely calibrated. We use this as a means to achieving multimodal output and not absolute calibration. Nevertheless, we still think that this does not decrease of the value of our approach as tackling absolute calibration is an orthogonal problem with readily available solutions  (Guo et al., 2017; Kull et al.,2019; Zhang et al., 2020).
> > > >
> > > > 7. Note: Point 7 here addresses the 9th point in the initial review.
> > > > We apologise for the confusion caused. We have now extended the description provided in Section 5.2.2 for the calibration experiment. We hope that this also clarifies it for other readers. Please let us know if you still have unanswered questions.

---

> ### Author Response · Authors · 2020-11-20
> **Added ECE score and reliability diagram**
>
> Please note that  we have now added the ECE score (2.15%) and a reliability diagram assessing the calibration network's quality, as suggested in your 8th point.

---

### Author Response · Authors · 2020-11-20
**Added reliability diagram and ECE score**

We would like to thank all reviewers for suggesting to improve our evaluation by using calibration-centric metrics. As per the request of Reviewers 2, 3, and 4 we constructed a reliability diagram (Fig. 6) and computed the ECE score (2.15%) for our calibration network used in the Cityscapes experiment, complementing the results we illustrate in Fig. 5. An extra paragraph has been added to the main text in Section 5.2.2 (page 9) discussing these results.

---

### Decision · Program_Chairs · 2021-01-07
**Final Decision**

**Decision:**

Reject

**Comment:**

This paper addresses stochastic semantic segmentation with a two-step approach: a standard segmentation network learned with cross-entropy serves as a guide to calibrate a second refinement network to generate diverse predictions while their expectation matches the calibration model.

The reviewers acknowledge the paper merits', e.g. the decoupling between the segmentation and generation networks. However, they also highlight serious concerns on the the clarity of the presentation, and the need for a consolidated evaluation.

The AC carefully reads the paper and the discussion among authors and reviewers. Despite improvements in paper presentation, the AC still considers that the paper would benefit from clarifications, e.g. the fact that the paper does not address calibration, and that stronger baselines as those mentioned by reviewers are needed for fully validating the approach.
Therefore, the AC recommends rejection.